# SpecExit: Accelerating Large Reasoning Model via Speculative Exit

**Rubing Yang** [* 1]  **Huajun Bai** [* 1 2]  **Song Liu** [1]  **Guanghua Yu** [† 1]  **Runzhi Fan** [1]  **Yanbin Dang** [1]  **Jiejing Zhang** [1]
**Kai Liu** [1]  **Jianchen Zhu** [1]  **Peng Chen** [1]

{rubingyang, lucayu}@tencent.com, bhj24@mails.tsinghua.edu.cn

## Abstract

Despite their strong performance on reasoning tasks, Large Reasoning Models (LRMs) often suffer from overthinking, producing unnecessarily long outputs and incurring high end-to-end latency, a significant limitation to their real-world deployment. To address overthinking, early-exit mechanisms have been proposed to terminate reasoning before typical completion, showing that this approach can effectively shorten generation length with minimal impact on accuracy. However, their reliance on probing mechanisms introduces a detection overhead that limits their end-to-end latency gains and compromises their generalizability across diverse problems. Inspired by the use of hidden states in speculative decoding, we propose **SpecExit**, a novel framework that predicts both future tokens and an early-exit signal directly from a lightweight draft model without probing overhead. Our method offers significant improvements, achieving up to 66% generation length reduction and 2.5× end-to-end speedup compared with the speculative decoding baseline, without compromising accuracy. Our method leverages the inherent signals from hidden states to provide effective early-exit signals, suggesting broader use of hidden states for efficient reasoning. Our code is available at: https://github.com/Tencent/AngelSlim.

## 1. Introduction

Large reasoning models (LRMs) such as OpenAI-o1 (OpenAI, 2024), DeepSeek-R1 (DeepSeek-AI et al., 2025) and Qwen (Qwen et al., 2025) have recently achieved state-of-the-art performance in complex tasks. These models follow the test-time scaling law (Snell et al., 2024; Brown et al., 2024; Muennighoff et al., 2025), where generating longer chain-of-thought (CoT) sequences (Wei et al., 2022) generally enhances model performance. However, this reliance on extended reasoning often leads to an *overthinking* problem, where models produce unnecessarily verbose outputs. This redundancy leads to both excessive token usage and high end-to-end latency, which limits LRMs' practical deployment.

To mitigate overthinking, researchers have proposed both inference-time and training-based strategies (Sui et al., 2025). Inference-time early-exit methods (Yang et al., 2025; Fu et al., 2024) rely on model-generated signals such as intermediate answers or output logits to terminate decoding once sufficient evidence is detected. These methods can shorten reasoning length without harming accuracy, but the probing overhead they introduce limits the actual latency gains. Training-based approaches, such as reinforcement learning (Aggarwal & Welleck, 2025; Yeo et al., 2025) and supervised fine-tuning (Ma et al., 2025b; Munkhbat et al., 2025), incur little runtime overhead during inference but risk altering the model's output distribution. As a result, existing methods struggle to deliver consistent improvements in end-to-end efficiency.

Speculative decoding (Chen et al., 2023; Leviathan et al., 2023) is a promising approach that improves efficiency without altering the target model's outputs. A lightweight draft model proposes multiple candidate tokens in advance, and the target model then verifies these candidates in parallel. This parallel verification allows the system to generate multiple tokens per forward pass of the target model, alleviating the inherent sequential bottleneck of autoregressive decoding and better utilizing modern GPU hardware. However, this strategy alone does not resolve the overthinking problem, as models still generate the full CoT. Recent advanced

---
[*]Equal contribution  [1]Tencent, Shenzhen, China [2]Department of Computer Science and Technology, Tsinghua University, Beijing, China.  Correspondence to: Guanghua Yu <lucayu@tencent.com>.

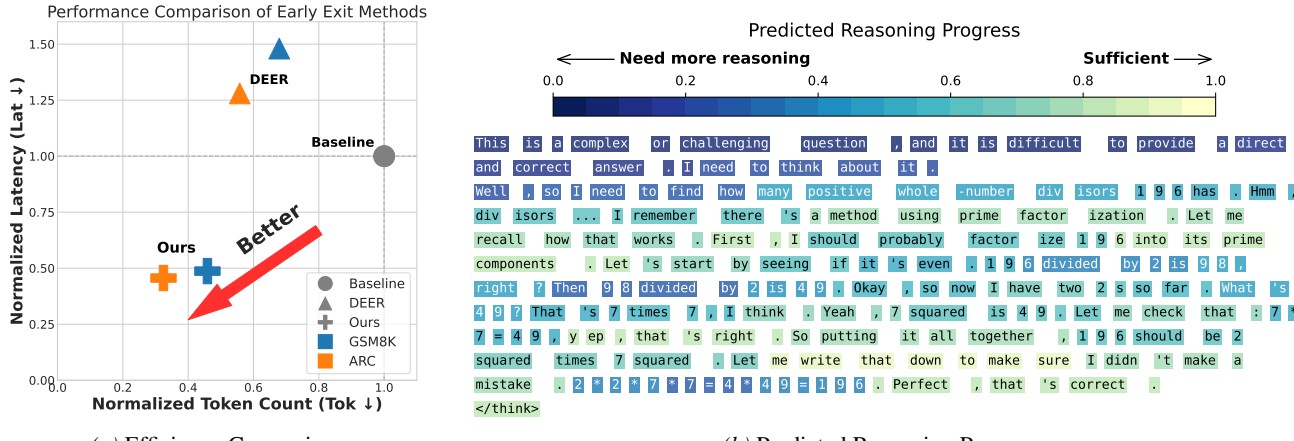

*(a) Efficiency Comparison*  *(b) Predicted Reasoning Progress.*

*Figure 1.* Effectiveness of the proposed method. (a) Statistical comparison showing that our approach produces shorter reasoning chains and faster inference than baselines. (b) visualizes the predicted reasoning progress on a MATH500 example, where darker colors denote insufficient reasoning and lighter colors denote sufficiency, demonstrating valuable signals can be extracted from hidden states regarding the model's reasoning process.

speculative decoding methods (Li et al., 2024a; Zhang et al., 2024) exploit hidden states to predict several future tokens. Other studies (Lin et al., 2025; Dong et al., 2025) also show that hidden states encode richer predictive signals beyond next-token probabilities. As shown in Figure 1b, the predicted reasoning progress, defined as a normalized value from 0 to 1 representing the fraction of effective reasoning completed relative to the shortest valid reasoning path, reveals that hidden states encode informative cues about reasoning sufficiency.

In this work, we introduce **SpecExit**, a reasoning-aware early-exit framework that leverages draft model hidden states not only to anticipate future tokens but also to predict early-exit signals. Unlike prior probing-based approaches, SpecExit requires no modifications to the target model and incurs no additional detection overhead. Instead, it extends the lightweight draft model with auxiliary prediction heads, enabling it to jointly output token distributions and reasoning-related signals in a single forward pass. By exploiting the latent information embedded in hidden states, SpecExit provides reliable criteria for dynamically terminating chain-of-thought generation when sufficient reasoning has been achieved.

We validate SpecExit on state-of-the-art reasoning models across mathematical, scientific, and logical benchmarks. For Qwen3-4B-Thinking-2507, DeepSeek-R1-Distill-Llama-8B, and Phi-4-reasoning models, SpecExit achieves up to 66% generation length reduction and 2.5× end-to-end latency speedup compared with the speculative decoding baseline. Our contributions can be summarized as follows:

- **Signals Extracted for Early-Exit**. We derive early-exit signals from hidden features and integrate them into spec-

ulative decoding, enabling reliable early-exit for efficient reasoning.

- **General and Practical Framework**. We implement SpecExit, a reasoning-aware early-exit framework, in both PyTorch and vLLM, making it easy to deploy across diverse inference environments.

- **Substantial End-to-End Performance Gains**. SpecExit reduces reasoning length by as much as 66% and delivers as high as 2.5x speedup in end-to-end inference compared with speculative decoding while maintaining accuracy.

**Conflict of Interest Disclosure.** The authors declare no financial conflicts of interest related to this work.

## 2. Motivation

Current early-exit methods face two challenges: runtime overhead from probing and limited generality from task-specific prompts. Since hidden states already reflect reasoning sufficiency, we propose leveraging them to replace costly probing for faster and more reliable inference across diverse tasks.

**Probing Overhead and Limited Generalizability.** Probing-based early-exit methods suffer from both latency overhead and poor generalizability across tasks. Shortened reasoning traces do not effectively translate into latency improvements. As shown in Figure 1a, DEER (Yang et al., 2025) reduces the generation length of GSM8K (Cobbe et al., 2021) and ARC-Challenge(Clark et al., 2018) by 32% and 44% respectively on Qwen3-4B-Thinking-2507, but even increases end-to-end latency than vanilla baseline. The gap arises because probing introduces extra computation, and its effectiveness is highly sensitive to both the task and

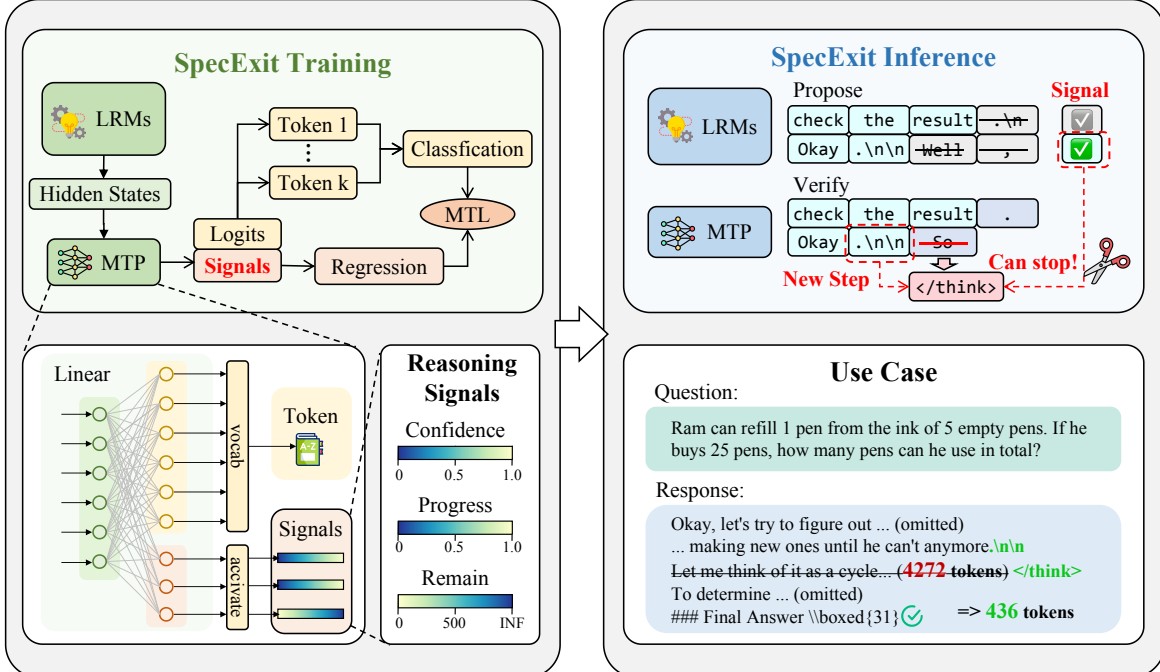

*Figure 2.* Overall architecture of the proposed SpecExit framework. The Multi-Token Prediction (MTP) layer is augmented to output both token logits and auxiliary signals. Training is performed with Multi-Task Learning (MTL), while at inference these signals guide speculative early stopping without modifying the backbone model. The example illustrates how redundant reasoning steps can be pruned while preserving final answer quality.

the model. A probing phrase like "Final Answer is" elicits effective intermediate answers in math problems, but fails in coding tasks where extra tokens are still generated. This drawback not only undermines generalizability across domains but also limits true latency savings.

**Signals from Hidden States**. Our preliminary experiments show that models' hidden states encode informative signals about its reasoning process. As illustrated in Figure 1b, we use a MLP trained on hidden states to predict reasoning progress. For complex tasks, the predicted signals appear darker at the beginning, reflecting the need for continued reasoning, but gradually shift to lighter colors as the model approaches a sufficient chain of thought. This progression suggests that hidden states provide fine-grained indicators of task complexity and reasoning sufficiency. Leveraging these internal signals offers an efficient alternative to costly probing, motivating our approach of utilizing signals from hidden states.

## 3. Method

### 3.1. Overall Architecture

**SpecExit Framework**. The overall design of this work aims to incorporate additional learnable signals into large model reasoning, thereby providing explicit decision-making criteria for early stopping. In the decoding process of large language models, hidden states not only encode semantic and contextual information for next-token prediction but also implicitly contain higher-level cues related to reasoning progress, generation quality, and content completeness. The Multi-Token Prediction (MTP) mechanism leverages these hidden states to project into the vocabulary space and predict multiple future tokens simultaneously, thereby improving inference efficiency. Inspired by this, we extend the MTP layer while keeping the backbone language model unchanged, introducing auxiliary prediction heads that allow the model to explicitly generate reasoning-related signals, including confidence, reasoning progress, and remaining reasoning length, alongside token distributions. This design preserves the original language modeling ability while providing learnable auxiliary variables for inference control, enabling efficient and dynamic reasoning regulation. The overall architecture of the SpecExit framework is shown in Figure 2.

**Model Structure**. We propose a speculative sampling-based early stopping mechanism for reasoning. In the MTP layer of the model, we extend the linear projection to include additional dimensions. In standard decoding, the hidden state $h \in \mathbb{R}^D$ is projected into the vocabulary space to predict the next token distribution, where $D$ is the hidden size of the model. In our method, the linear layer output is

extended as:

$$Wh = \begin{bmatrix} W_{\text{tok}}h, \ W_{\text{conf}}h, \ W_{\text{prog}}h, \ W_{\text{rem}}h \end{bmatrix}, \qquad (1)$$

where $W_{\text{tok}}h$ produces standard token predictions, while $W_{\text{conf}}h, W_{\text{prog}}h, W_{\text{rem}}h$ predict confidence, reasoning progress, and remaining reasoning length, respectively. These additional signals serve as observable indicators for deciding whether to stop reasoning early, thus reducing redundant computation without compromising output quality.

### 3.2. Signal-Extracted Training

**Data Construction**. We first obtain the complete response generated by the base language model and extract the reasoning content enclosed within the `<think>` and `</think>` tokens. To identify the effective reasoning trace, we iteratively attempt to insert the closing marker `</think>` after each paragraph and verify whether the resulting final answer matches the original output. If the answer remains consistent, the subsequent reasoning content is regarded as redundant. Consequently, only the minimal reasoning segment required to produce the correct answer is retained as training data. For code generation tasks, the shortest valid reasoning path is determined by inserting the `</think>` marker at successive paragraph boundaries and verifying whether the generated code passes the full test suite. The earliest insertion point that still produces a passing solution defines the minimal reasoning path.

**Signal Annotation**. CONFIDENCE is defined as the geometric mean of the logit probabilities across prediction steps, reflecting the reliability of the generation; REMAINING reasoning length is defined as the remaining number of reasoning tokens from the initial `<think>` marker to the earliest valid insertion point of `</think>` that still yields the correct answer; PROGRESS is represented as a normalized value increasing from 0 to 1, capturing the relative progression of the reasoning CoT.

**Signal Regression**. We propose a cost-efficient extension by introducing a small number of additional dimensions into the linear projection layer of the MTP module for regressing reasoning signals. These dimensions are orthogonal to the vocabulary classification weights, ensuring that signal regression does not interfere with the convergence of speculative decoding training.

The Multi-Task Learning (MTL) overall training objective jointly optimizes token classification and signal regression, defined as:

$$\mathcal{L} = \mathcal{L}_{\text{cls}} + \lambda_c \mathcal{L}_{\text{conf}} + \lambda_p \mathcal{L}_{\text{prog}} + \lambda_r \mathcal{L}_{\text{rem}}, \qquad (2)$$

where $\mathcal{L}_{\text{cls}}$ is the standard cross-entropy loss for vocabulary

prediction, and $\mathcal{L}_{\text{conf}}, \mathcal{L}_{\text{prog}}, \mathcal{L}_{\text{rem}}$ correspond to the regression losses for confidence, progress, and remaining reasoning length, with $\lambda_c, \lambda_p, \lambda_r$ denoting dynamic weighting coefficients. Specifically:

Confidence and progress are optimized using mean squared error (MSE), remaining reasoning length is optimized with mean squared logarithmic error (MSLE):

$$\mathcal{L}_{\text{conf}} = \frac{1}{N} \sum_{i=1}^{N} (\text{sigmoid}(\hat{c}_i) - c_i)^2, \qquad (3)$$

$$\mathcal{L}_{\text{prog}} = \frac{1}{N} \sum_{i=1}^{N} (\text{sigmoid}(\hat{p}_i) - p_i)^2, \qquad (4)$$

$$\mathcal{L}_{\text{rem}} = \frac{1}{N} \sum_{i=1}^{N} \left( \hat{r}_i - \log(1 + r_i) \right)^2, \qquad (5)$$

where $\hat{c}_i$, $\hat{p}_i$, and $\hat{r}_i$ represent the model-predicted confidence, progress, and remaining reasoning length, respectively; $c_i$, $p_i$, and $r_i$ denote their corresponding ground-truth values; and $N$ is the total number of samples.

**Dynamic Weighting**. Since the regression losses of signals converge faster than the token classification loss, we adopt a gradient-based dynamic weighting strategy to balance the contributions of different tasks. This mechanism assigns higher weights to tasks with smaller gradient magnitudes, preventing tasks with larger gradients from dominating the learning process and ensuring all tasks are effectively optimized. Formally, the mechanism is defined as:

$$\lambda_j = \frac{\|\nabla_\theta \mathcal{L}_j\|^{-1}}{\sum_k \|\nabla_\theta \mathcal{L}_k\|^{-1}}, \quad \mathcal{L}_{\text{total}} = \sum_j \lambda_j \mathcal{L}_j, \qquad (6)$$

where $\mathcal{L}_j \in \{\mathcal{L}_{\text{cls}}, \mathcal{L}_{\text{conf}}, \mathcal{L}_{\text{prog}}, \mathcal{L}_{\text{rem}}\}$, $\nabla_\theta \mathcal{L}_j$ denotes the gradient of task $j$ with respect to the model parameters, and $\lambda_j$ is the dynamically computed weight. This formulation ensures that the gradient contributions of all tasks are balanced, which facilitates stable convergence in multi-task optimization, as shown in Figure 3.

### 3.3. Signal-Guided Inference

**Overall Procedure**. To assess the feasibility and effectiveness of the proposed signal-guided inference, we extend the speculative decoding framework and implement the inference pipeline in both **PyTorch** and **vLLM**. The key modification is introduced in the forward pass of the target model. In addition to computing the logits for next-token prediction, we extract the hidden states of the last accepted token as an intermediate representation. This representation is mapped by a lightweight linear module to

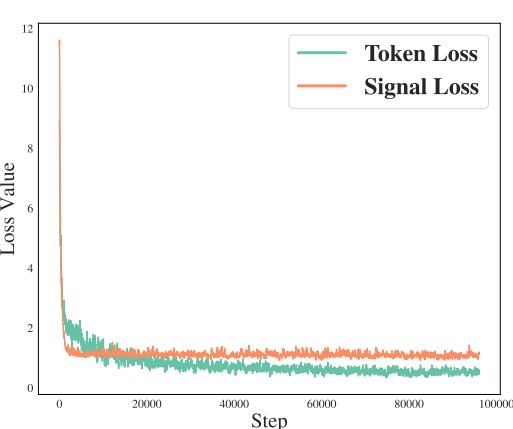

*Figure 3.* Convergence of token classification loss and signal regression loss during training.

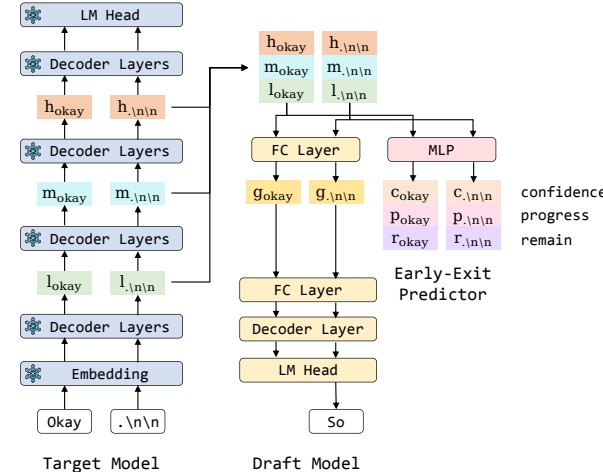

*Figure 4.* Overview of the inference pipeline integrating EA-GLE3 (Li et al., 2025) with the proposed SpecExit mechanism.

reasoning-related signals: confidence, progress, and remaining reasoning length. These signals jointly inform the reasoning early-exit criterion, enabling adaptive control of the inference process. Figure 4 illustrates the speculative decoding pipeline augmented with the proposed signal-guided early-exit mechanism.

---

**Algorithm 1** Inference procedure with signal-guided speculative exit

---

**Input:** Draft model $M_d$, target model $M_t$, tokenizer, thresholds
**Output:** Generated sequence $y$
$is\_thinking \leftarrow$ true
$step\_split\_tokens \leftarrow \{$ids of "\n\n", ".\n\n", ...$\}$
$stop\_think\_token \leftarrow$ id of $</think>$
**while** not terminated **do**
    Extract hidden state of $t_{acpt}$ and generate candidate tokens with $M_d$
    Compute $signals$ (confidence, progress, remaining)
    Update smoothed $signals$
    Concatenate last accepted token with draft candidates
    Forward through $M_t$ with tree attention
    Accept tokens $t_{acpt}$, accept length $l_{acpt}$, recover token $t_{rec}$
    **if** $is\_thinking$ **and** $\exists\, t_{acpt} \in step\_split\_tokens$ **and** signals exceed thresholds **then**
        Set $l_{acpt}$ to corresponding step-split position
        Set $t_{rec} \leftarrow stop\_think\_token$
        Update KV-cache and hidden states
        $is\_thinking \leftarrow$ false
    **end if**
**end while**

---

**Speculative Decoding Inference Procedure.** In specu-

lative decoding, the draft model proposes a sequence of draft tokens, which are then verified by the target model. Only the verified tokens are committed to the target model's KV-cache. The target model subsequently generates a recover token following the last accepted draft token. The hidden states produced during verification are fed back to the draft model to guide the next draft generation. This process iterates, with both recover tokens and new draft tokens undergoing further verification.

**Stopping Conditions**. To ensure that early-exit decisions occur at semantically coherent boundaries, we introduce a class of special markers called *step split tokens*, which indicate natural segmentation points in the generated text. Specifically, step split tokens can be divided into two categories: PARAGRAPH DELIMITERS (e.g., . \n\n), which mark the end of a paragraph or reasoning unit, and DISCOURSE MARKERS (e.g., "Wait", "But", or "Therefore"), which often signal semantic transitions or logical shifts during reasoning. Since the segmentation strategy based on PARAGRAPH DELIMITERS is more general, this strategy is adopted by default in subsequent experiments. Examples of commonly observed discourse markers in reasoning traces are shown in Figure 11 (Appendix). When a sampled token belongs to the above set, the early-exit logic is triggered. If the smoothed signal exceeds the predefined threshold, the system determines that the reasoning process has been sufficiently explored. The complete inference process is summarized in Algorithm 1. In this case, the accepted output length is truncated at the position of the step split token, and the target model's recover token is replaced with a special reasoning-end marker (e.g., </think>), thereby ensuring that the termination point lies at a natural boundary while maintaining coherence of the generated text.

**Signal Smoothing**. Since raw signals may exhibit signif-

*Table 1.* Performance comparison of various reasoning methods on mathematical, scientific, general, and coding benchmarks. "Acc" denotes accuracy, "Tok" denotes token count, and "Lat" denotes total end-to-end latency. ↑ indicates that higher values are better, while ↓ indicates that lower values are better. For the early-exit methods (NoThink, DEER, and SpecExit*), the highest and second-highest Acc values are marked in **bold** and underline, respectively. Across all methods, the smallest and second-smallest Tok and Lat values are marked **bold** and underline, respectively. SpecExit* uses default parameter settings consistent with the best variants in ablation studies.

| Method | Math | | | | | | | | | Coding | | | Science | | | Logic | | | Average | | |
|---|---|---|---|---|---|---|---|---|---|---|---|---|---|---|---|---|---|---|---|---|---|
| | GSM8K | | | MATH500 | | | AIME | | | HUMANEVAL+ | | | GPQA-D | | | ARC-Challenge | | | | | |
| | Acc↑ | Tok↓ | Lat↓ | Acc↑ | Tok↓ | Lat↓ | Acc↑ | Tok↓ | Lat↓ | Acc↑ | Tok↓ | Lat↓ | Acc↑ | Tok↓ | Lat↓ | Acc↑ | Tok↓ | Lat↓ | Acc↑ | Tok↓ | Lat↓ |
| *Qwen3-4B-Thinking-2507* | | | | | | | | | | | | | | | | | | | | | |
| *Vanilla* | 95.3 | 1414 | 155.6 | 96.6 | 6719 | 530.1 | 86.7 | 19577 | 243.3 | 90.9 | 5079 | 175.3 | 68.7 | 9041 | 325.8 | 95.6 | 1812 | 156.5 | 89.0 | 7274 | 264.4 |
| *NoThink* | **95.2** | 1631 | 204.2 | 96.6 | 6395 | 488.5 | 86.7 | 19816 | 243.2 | 88.4 | 4480 | 131.5 | 67.2 | 8833 | 276.8 | **95.1** | 1889 | 159.8 | 88.2 | 7174 | 250.7 |
| *DEER* | 94.3 | 960 | 230.3 | 94.4 | 4893 | 519.6 | 70.0 | **17838** | 218.6 | 86.6 | **4079** | 242.4 | 67.2 | 9053 | 505.2 | 94.6 | 1011 | 200.3 | 84.5 | 6306 | 319.0 |
| *EAGLE3* | 94.8 | 1408 | 140.3 | 96.6 | 6670 | 395.7 | 80.0 | 19792 | 206.1 | 87.2 | 5178 | 81.7 | 67.7 | 8975 | 212.2 | 95.7 | 1822 | 164.2 | 87.0 | 7308 | 200.0 |
| *SpecExit\** | 93.8 | **649** | **75.8** | **96.8** | 4777 | 367.9 | **90.0** | 17769 | 187.3 | **89.6** | 4319 | **58.4** | **68.7** | **7011** | **137.0** | 94.5 | **588** | **71.4** | **88.9** | **5852** | **149.6** |
| *DeepSeek-R1-Distill-Llama-8B* | | | | | | | | | | | | | | | | | | | | | |
| *Vanilla* | 77.3 | 1008 | 629.4 | 81.8 | 6878 | 857.1 | 36.7 | 22170 | 307.0 | 74.4 | 6287 | 445.5 | 43.6 | 8857 | 574.0 | 49.9 | 1917 | 628.5 | 60.5 | 7853 | 573.6 |
| *NoThink* | 54.6 | **233** | **22.2** | 55.2 | **1643** | **262.8** | 10.0 | 8744 | 184.1 | 46.3 | **472** | **7.3** | 26.8 | **1200** | **166.6** | 12.6 | **135** | **13.6** | 34.3 | **2071** | **109.4** |
| *DEER* | 74.7 | 710 | 484.8 | **80.8** | 3533 | 973.3 | **40.0** | 15619 | 272.3 | 79.3 | 4206 | 269.2 | 40.9 | 8492 | 521.5 | 47.5 | 1029 | 531.3 | 60.5 | 5598 | 508.7 |
| *EAGLE3* | 76.6 | 976 | 276.9 | 80.8 | 6172 | 593.6 | 30.0 | 25686 | 228.1 | 78.7 | 5312 | 346.5 | 43.9 | 8749 | 420.1 | 59.2 | 1378 | 496.4 | 62.0 | 8046 | 393.6 |
| *SpecExit\** | **74.6** | 333 | 112.6 | 80.6 | 1968 | 348.3 | 36.7 | **8160** | **176.0** | **81.7** | 3105 | 118.1 | **46.0** | 6849 | 307.5 | **50.3** | 500 | 253.7 | **61.8** | 3486 | 219.4 |
| *Phi-4-reasoning* | | | | | | | | | | | | | | | | | | | | | |
| *Vanilla* | 95.8 | 709 | 207.2 | 94.9 | 2122 | 543.7 | 74.2 | 10980 | 536.3 | 72.6 | 2059 | 300.2 | 68.7 | 7544 | 726.7 | 96.6 | 607 | 193.2 | 83.8 | 4003 | 417.9 |
| *NoThink* | 95.7 | 668 | 197.3 | **94.1** | 2051 | 554.8 | 70.7 | 11104 | 509.4 | **72.8** | 1919 | 297.3 | 64.7 | 7334 | 710.0 | **96.7** | 588 | 178.0 | 82.5 | 3950 | 405.7 |
| *DEER* | 95.5 | 582 | 223.5 | 92.4 | **1502** | 516.0 | 60.0 | **7003** | 507.4 | 66.3 | **1420** | 211.1 | 65.2 | **4479** | 1296.1 | 96.0 | 540 | 183.4 | 79.3 | **2584** | 481.2 |
| *EAGLE3* | 95.2 | 707 | 153.1 | 94.5 | 2136 | 324.2 | 74.0 | 10657 | 308.5 | 72.0 | 2035 | 155.8 | 68.2 | 7512 | 478.2 | 96.7 | 615 | 128.9 | 83.2 | 3932 | 258.1 |
| *SpecExit\** | **95.8** | **400** | **61.0** | 93.6 | 1750 | **271.9** | **74.7** | 9988 | 272.6 | 72.0 | 1605 | **131.8** | **67.7** | 6922 | 422.9 | 95.8 | **286** | **80.0** | **83.3** | 3579 | **208.4** |

icant volatility, relying directly on them risks premature or unstable termination. To enhance robustness, we apply an Exponentially Weighted Moving Average (EWMA) to smooth the signals across steps. At each iteration, the smoothed value is updated as a weighted average of the current raw signal and the previous smoothed value, with the smoothing factor controlling the balance between recent and past observations. A smaller factor emphasizes historical stability, yielding smoother traces that are less sensitive to transient noise. This ensures that termination decisions reflect consistent trends rather than isolated fluctuations.

## 4. Experiments

### 4.1. Experimental Setup

To evaluate the effectiveness of our SpecExit framework, we conducted a comprehensive set of experiments across multiple domains. Specifically, we used the **GSM8K** (Cobbe et al., 2021), **MATH500** (Hendrycks et al., 2021) and **AIME** (MAA Committees) datasets for mathematical reasoning, the **HumanEvalPlus** (Liu et al., 2023) dataset for coding, the **GPQA Diamond** (Rein et al., 2024) dataset for science, and the **ARC-Challenge** (Clark et al., 2018) dataset for logic. Experiments are conducted on three mainstream LRMs: **Qwen3-4B-Thinking-2507** (Qwen et al., 2025),

**DeepSeek-R1-Distilled-Llama-8B** (DeepSeek-AI et al., 2025) and **Phi-4-reasoning** (Abdin et al., 2025).

We compare our SpecExit method against several baselines: **Vanilla**, which represents full generation without any early-exit mechanism; **NoThink** (Ma et al., 2025a), which skips the reasoning phase; **DEER** (Yang et al., 2025), a dynamic early-exit method; and **EAGLE3** (Li et al., 2025), a speculative decoding baseline. For a fair and consistent comparison, the speculative decoding component in our system adopts the same draft-model architecture as EAGLE3, namely a one-layer causal model whose hidden size matches the corresponding target model. The draft model is trained together with SpecExit signals using the same training procedure to ensure comparable conditions.

Our performance analysis is based on three key metrics, as detailed in Table 1: **Accuracy** (↑), **Token** (↓) count and end-to-end **Latency** (↓). All experimental results are obtained by implementing our early-exit strategy in vLLM (woo), and running inference on an 8×H20 GPU cluster.

### 4.2. Main Results

We first evaluate the proposed SpecExit against baseline reasoning approaches on mathematical, scientific, coding, and logical benchmarks. As shown in Table 1, SpecExit

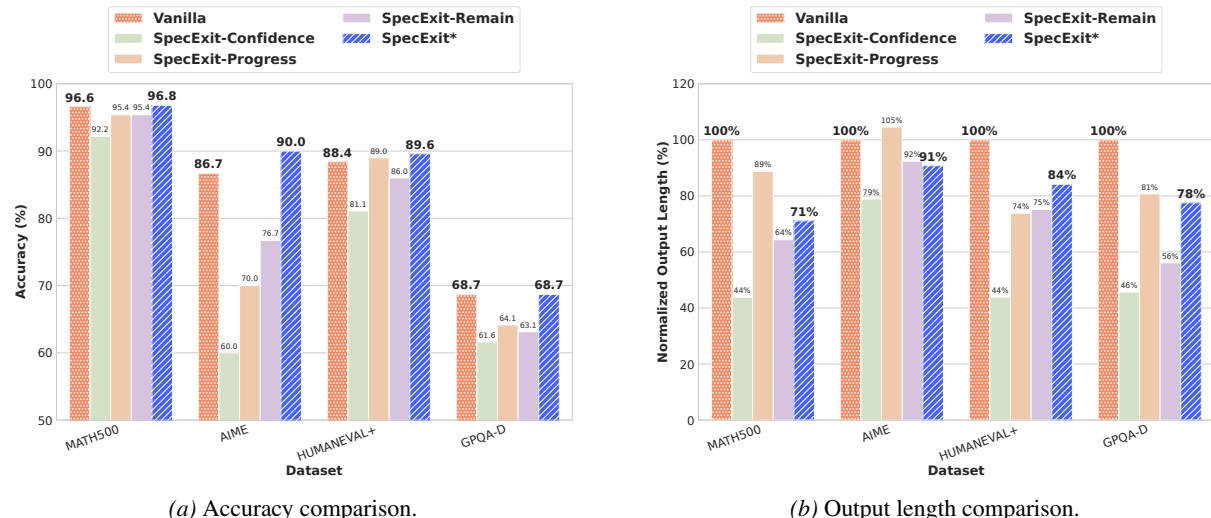

*(a)* Accuracy comparison.

*(b)* Output length comparison.

*Figure 5.* Ablation study of SpecExit signal types on Qwen3-4B-Thinking-2507.

*Table 2.* Ablation study of different signal smoothing methods on Qwen3-4B-Thinking-2507. The highest Acc values are marked in bold for the early-exit methods.

| Method | MATH500 | | AIME | | HUMANEVAL+ | | GPQA-D | | Average | |
|---|---|---|---|---|---|---|---|---|---|---|
| | Acc | Tok | Acc | Tok | Acc | Tok | Acc | Tok | Acc | Tok |
| *Vanilla* | 96.6 | 6719 | 86.7 | 19577 | 88.4 | 5133 | 68.7 | 9041 | 85.1 | 10118 |
| *NoSmooth* | 94.2 | 3608 | 73.3 | 17832 | 92.1 | 2789 | 62.1 | 4066 | 80.4 | 7074 |
| *Momentum* | 91.8 | 2230 | 60.0 | 12427 | 83.5 | 2219 | 64.7 | 3406 | 75.0 | 5071 |
| *Sliding Window* | 95.2 | 4444 | 80.0 | 19184 | 86.6 | 4342 | 62.1 | 4738 | 81.0 | 8177 |
| *Paragraph Mean* | 95.4 | 4285 | 76.7 | 18231 | 84.8 | 4569 | 65.7 | 4726 | 80.6 | 7953 |
| *SpecExit*(EWMA)* | **96.8** | 4777 | **90.0** | 17769 | **89.6** | 4319 | **68.7** | 7011 | **86.3** | 8469 |

consistently achieves substantial reductions in both output length and inference latency while maintaining comparable or even higher accuracy.

Across benchmarks, SpecExit significantly shortens reasoning traces, with up to 54% and 53% reduction on GSM8K and ARC-Challenge for Qwen3-4B-Thinking-2507, and up to 66% and 64% reduction for DeepSeek-R1-Distill-Llama-8B. The reduced reasoning length corresponds to measurable efficiency improvements: SpecExit achieves a up to 1.9x latency reduction with Qwen3-4B-Thinking-2507 and up to 2.5x speedup with DeepSeek-R1-Distill-Llama-8B on GSM8K, compared with the speculative decoding baseline EAGLE3. Importantly, these gains come only with marginal accuracy differences, confirming that early termination of redundant reasoning does not harm task performance. By contrast, prior inference-time methods primarily focus on reducing output length, but the latency gains they achieve are relatively modest. In some datasets, the additional computational overhead even leads to slower inference than the standard think mode. Notably, for Qwen3-4B-Thinking-

2507 and Phi-4-reasoning models, inserting the `</think>` token at the beginning of reasoning in the NoThink baseline still fails to suppress reasoning, yielding output lengths similar to the vanilla Think mode and occasionally slightly longer.

Overall, these results demonstrate that SpecExit achieves a favorable balance between efficiency and accuracy, highlighting the practicality of integrating reasoning-aware early-exit strategies into LRMs inference.

### 4.3. Ablation Study

**Signal Type**. To investigate the impact of individual reasoning signals in SpecExit, we conduct ablation studies on confidence, progress, and remaining token length, along with a combined configuration (SpecExit*) that integrates all three. As shown in Figure 5, the confidence-only variant yields the largest token reduction but overestimates the model's certainty, resulting in noticeable accuracy drops on complex benchmarks. The predicted reasoning progress increases

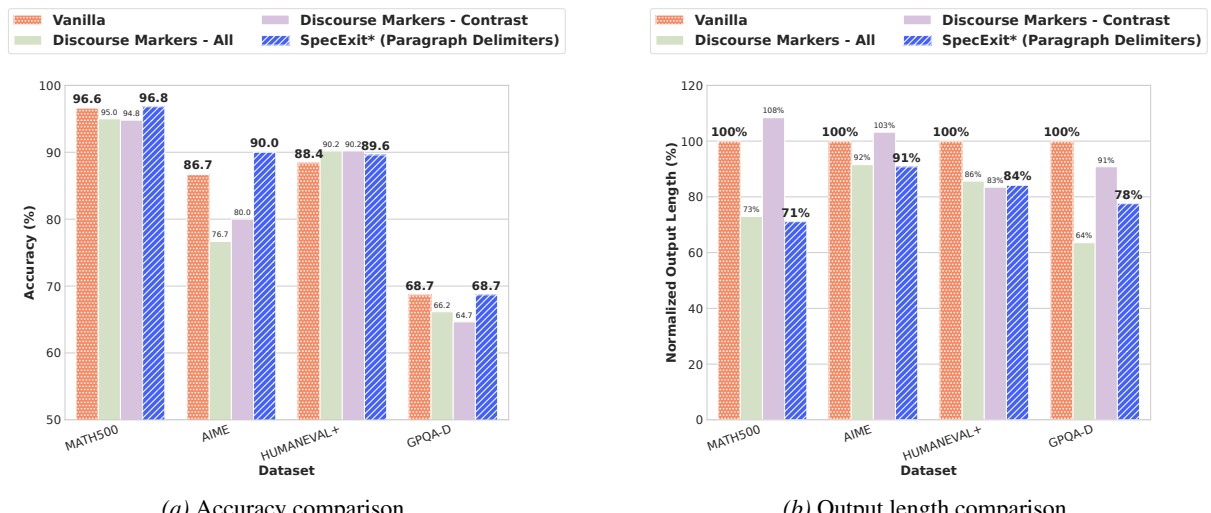

*(a)* Accuracy comparison.        *(b)* Output length comparison.

*Figure 6.* Ablation study of step split tokens strategies on Qwen3-4B-Thinking-2507.

sharply in the early steps yet continues to fluctuate during iterative reflection. Remaining token length is generally high at the beginning of inference but often triggers premature exits on complicated problems. By integrating all signals, SpecExit* leverages their complementary strengths, preserving competitive accuracy while substantially reducing tokens, demonstrating that multi-signal integration mitigates individual biases and enables more reliable early stopping.

**Signal Smoothing**. In order to investigate the influence of different smoothing strategies on the stability and performance of early-exit decisions, we conducted a series of ablation experiments comparing multiple approaches. As shown in Table 2, removing smoothing increases the volatility of cognitive signals, leading to inconsistent early exits and increased token consumption. The momentum-based prediction strategy significantly reduces token usage, though it may slightly degrade accuracy due to overly aggressive early termination. Smoothing using sliding-window and paragraph-level averaging offers a better trade-off, maintaining accuracy while improving efficiency. Among all methods, Exponential Weighted Moving Average (EWMA) strikes the most consistent balance, providing both stability and reliability. These results demonstrate that appropriate smoothing is essential for reliable early-exit behavior, as it mitigates the influence of transient fluctuations in the raw cognitive signals.

**Step Split Tokens**. To evaluate the influence of different step split strategies on early-exit performance, we conducted ablation experiments comparing paragraph delimiters, general discourse markers, and a contrastive subset of discourse markers. Discourse markers indicate semantic transitions or reasoning shifts, but are dependent on the underlying data and model, limiting their generality. In prior work

on dynamic early-exit methods, contrastive subsets of discourse markers (e.g., "Wait", "But", "Alternatively") are frequently used to capture reasoning-relevant transitions. In contrast, paragraph delimiters ($\n\n$) provide a more general segmentation that does not rely on model-specific or dataset-specific patterns. As shown in Figure 6, using paragraph delimiters achieves competitive accuracy and token reduction, demonstrating that a general segmentation strategy can be effective for early-exit decisions while maintaining coherence in reasoning traces.

In summary, our ablation studies on signal types, smoothing strategies, and step split methods provide key insights for improving early-exit decision-making. The integration of multiple signals strikes the best balance between accuracy and token efficiency, while appropriate smoothing methods stabilize cognitive signals and enhance the consistency of early exits. Additionally, using general segmentation strategies, such as paragraph delimiters, improves the generalizability of early-exit systems across diverse datasets. These findings emphasize the importance of a holistic approach, where complementary strategies jointly enhance both efficiency and reliability.

## 5. Related Work

**Efficient Reasoning.** To mitigate unnecessary CoT generation in LRMs (Chen et al., 2024; Sui et al., 2025), prior work has explored both training-based and inference-time strategies. Training-based approaches typically modify model behavior through reinforcement learning with length-sensitive objectives (Aggarwal & Welleck, 2025; Yeo et al., 2025; Shen et al., 2025; Huang et al., 2026) or supervised fine-tuning on reasoning traces of varying lengths (Ma et al., 2025b; Munkhbat et al., 2025). While effective in shorten-

ing outputs, these methods demand substantial retraining cost and can distort the model's output distribution, raising concerns about reliability and generalization to unseen tasks. Inference-time methods avoid retraining and instead attempt to stop reasoning dynamically by monitoring model signals such as logits (Yang et al., 2025) or intermediate answers (Fu et al., 2024). Although these methods show that early stopping can reduce reasoning length without degrading accuracy, their reliance on probing introduces additional computation and often emphasizes token count reduction rather than true end-to-end latency improvements.

**Speculative Decoding and Hidden States.** Speculative decoding (Chen et al., 2023; Leviathan et al., 2023) is a widely adopted technique for accelerating decoding speed, where a lightweight draft model proposes candidate tokens that a larger target model verifies in a single pass. Recent methods (Cai et al., 2024; Li et al., 2024a;b; 2025; Zhang et al., 2024) leverage hidden states to predict multiple future tokens. Beyond speculative decoding, several studies (Lin et al., 2025; Zhang et al., 2025b; Dong et al., 2025; Zhang et al., 2025a), have revealed that hidden states contain broader information about future outputs, including correctness, response length, and reasoning paths. Building on this insight, our method extends speculative decoding by training hidden states not only to forecast future tokens but also to produce an early-exit signal.

## 6. Conclusion

In this work, we propose **SpecExit**, a reasoning-aware early-exit framework that leverages latent signals from models' hidden states to dynamically terminate reasoning processes in LRMs. By concatenating auxiliary prediction heads to a lightweight draft model, SpecExit simultaneously predicts future tokens and early-exit signals in a single forward pass, eliminating the probing overhead required by previous approaches. Our experiments across diverse tasks and models demonstrate that SpecExit substantially reduces reasoning length by up to 66% and achieves significant end-to-end latency improvements up to 2.5x without compromising accuracy. The proposed method highlights the potential of hidden states as informative signals for efficient reasoning and establishes a practical pathway for deploying LRMs in real-world scenarios.

## 7. Limitations

The effectiveness of SpecExit depends on the presence of redundant reasoning in the base model's output. For tasks that approach the model's capability boundary, most generated tokens contribute meaningfully to the final answer, leaving limited room for early exit without degrading accuracy.

The construction of training data for the exit-signal heads

requires verifiable outputs with deterministic correctness criteria. This requirement restricts the direct applicability of the current framework to open-ended generation or dialogue tasks. Extending SpecExit to such settings would necessitate alternative supervision strategies, for example reward model scores or semantic similarity metrics.

The signal prediction heads are trained on data drawn from specific domains. When the model encounters out-of-distribution tasks, prediction errors in the exit signals may reduce the reliability of early-exit decisions, potentially causing premature termination or insufficient token reduction.

Deploying SpecExit in production serving frameworks other than vLLM may require additional engineering effort to accommodate differences in scheduling policies and memory management mechanisms across systems.

## Impact Statement

This paper presents work whose goal is to advance the field of machine learning. There are many potential societal consequences of our work, none of which we feel must be specifically highlighted here.

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

# A. Appendix

## A.1. Supplementary Experiments

**Analysis of Hidden States:** To investigate whether intermediate hidden representations encode discriminative signals relevant to reasoning sufficiency, we analyze hidden states extracted from multiple depth stages of the reasoning traces. We apply Principal Component Analysis (PCA) and Uniform Manifold Approximation and Projection (UMAP) to project the hidden states into lower-dimensional spaces and analyze their spatial distribution. The analysis reveals that, across datasets of varying difficulty, the hidden states at the "start," "mid," and "end" positions exhibit clear clustering patterns. Moreover, the hidden states at the start of the reasoning trace show notable cross-dataset similarity, as illustrated in Figure 7a. In addition, when examining the relationship between the embedding representations of hidden states and the paragraph geometric mean of probabilities, we observe that hidden states also form meaningful clusters under different probability ranges, as illustrated in Figure 7b. These observations suggest that hidden representations indeed capture information related to dataset difficulty, reasoning progress, paragraph-level average probability, thereby providing preliminary evidence that functions of intermediate hidden states can serve as reliable proxies for reasoning sufficiency.

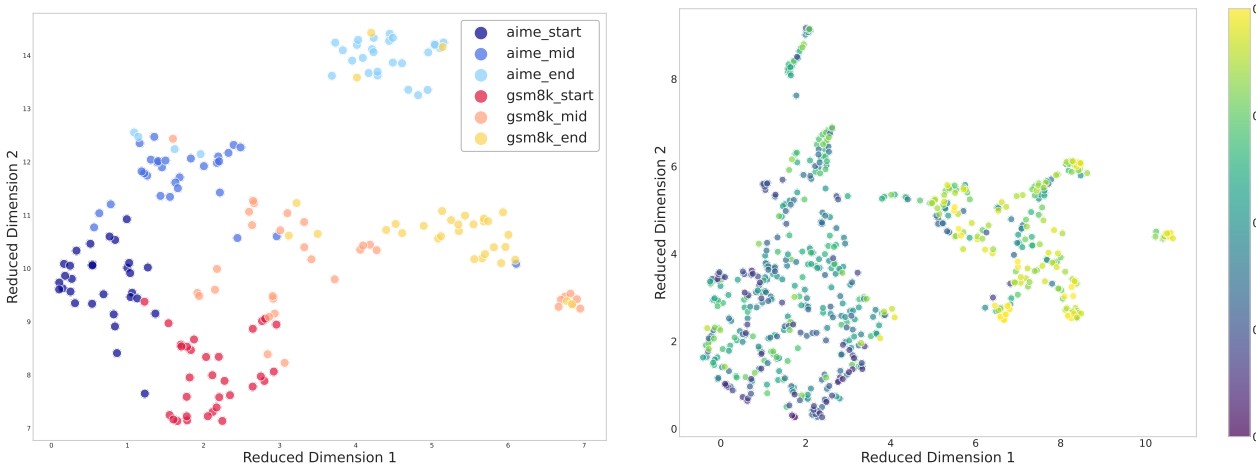

*(a)* Distribution of hidden states on the GSM8K and AIME datasets. The labels "start," "mid," and "end" correspond to early, intermediate, and final positions along the reasoning trace, respectively.

*(b)* Relationship between the embedding representations of hidden states and the paragraph geometric mean of probabilities, showing clustering patterns under different confidence ranges.

*Figure 7.* Analysis of hidden-state representations on the GSM8K and AIME datasets. (a) shows the distribution of hidden states along different reasoning positions; (b) shows the relationship between hidden states and paragraph-level confidence signals.

**Threshold Calibration:** In the ablation study of SpecExit signal types, the following thresholds are applied as stopping conditions for the respective signal types: SpecExit-Confidence requires predicted confidence value greater than 0.9, SpecExit-Progress requires predicted progress value greater than 0.8, SpecExit-Remain requires predicted remaining reasoning length value less than 100, and SpecExit* combines thresholds with a predicted confidence value greater than 0.8, predicted progress greater than 0.3, and predicted remaining reasoning length less than 200.

In addition to the thresholds reported above, we provide two complementary procedures that further calibrate the stopping criteria using a small held-out calibration set. Specifically, we sample 90 instances of varying difficulty from the validation split of training data and conduct the following analyses:

(1) Statistical distribution-based thresholding. During data construction, we have access to the shortest valid reasoning path for each problem, and thus we can determine whether stopping at the end of any intermediate paragraph would still yield a correct final answer. By examining the empirical distribution of the predicted signals at these paragraph boundaries and correlating them with correctness, we obtain the distributions shown in Figure 9. These distributions allow us to derive a signal-specific threshold that maximizes the retention of correct answers under early stopping.

(2) Design space exploration over the threshold search space. We additionally perform a design space exploration (DSE) over a predefined grid of confidence, progress, and remaining-length thresholds. Using vLLM to run this evaluation pipeline on the 90-sample calibration set, the full search requires approximately 2.5 hours on an 8×H20 GPU cluster. Among

all threshold combinations, we select those lying on the Pareto frontier that best trade off accuracy preservation against reduction in reasoning length. The results are shown in Figure 9.

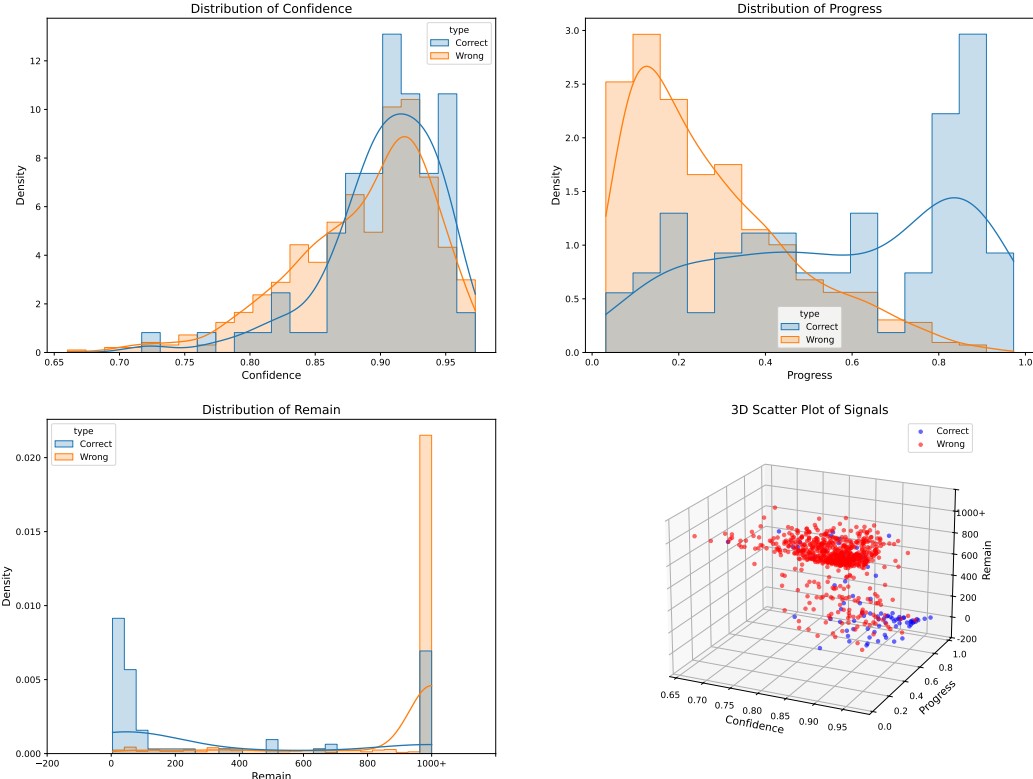

*Figure 8.* Distribution of signals with correct and wrong answers in calibration data.

## A.2. Implementation Details

**Model Architectures:** We evaluate SpecExit on three large reasoning models (LRMs): **Qwen3-4B-Thinking-2507** (Qwen et al., 2025), **DeepSeek-R1-Distilled-Llama-8B** (DeepSeek-AI et al., 2025) and **Phi-4-reasoning** (Abdin et al., 2025). For speculative decoding, the draft models adopt the EAGLE3 (Li et al., 2025) architecture. The draft models are single-layer causal models whose hidden sizes match those of the corresponding target models. The input embedding layer of each draft model is shared with its corresponding target model to ensure tokenizer compatibility, while the output head uses a compact vocabulary of 32k high-frequency tokens. The detailed configurations are summarized in Table 3.

Figure 4 and Figure 10 provides an overview of how SpecExit integrates with the EAGLE3 speculative decoding pipeline. During inference, the draft model produces multi-level token predictions, and simultaneously, our early-exit module tracks the evolution of three reasoning-related signals to determine whether the target model can safely terminate the thinking phase. The design is architecture-agnostic and can be combined with other multi-token-prediction frameworks such as Medusa. In practical deployment, the fully-connected layer of the MTP head (e.g., EAGLE3 or Medusa) can be fused with the early-exit module, effectively hiding additional operator invocation latency.

*Table 3.* Target and draft model configurations.

| Model Name | Role | Architecture | Hidden Size | Layers | Vocab Size |
|---|---|---|---|---|---|
| Qwen3-4B-Thinking-2507 | Target | `Qwen3ForCausalLM` | 2560 | 36 | 151936 |
| | Draft | `Eagle3LlamaForCausalLM` | 2560 | 1 | 32000 |
| DeepSeek-R1-Distilled-Llama-8B | Target | `LlamaForCausalLM` | 4096 | 32 | 128256 |
| | Draft | `Eagle3LlamaForCausalLM` | 4096 | 1 | 32000 |

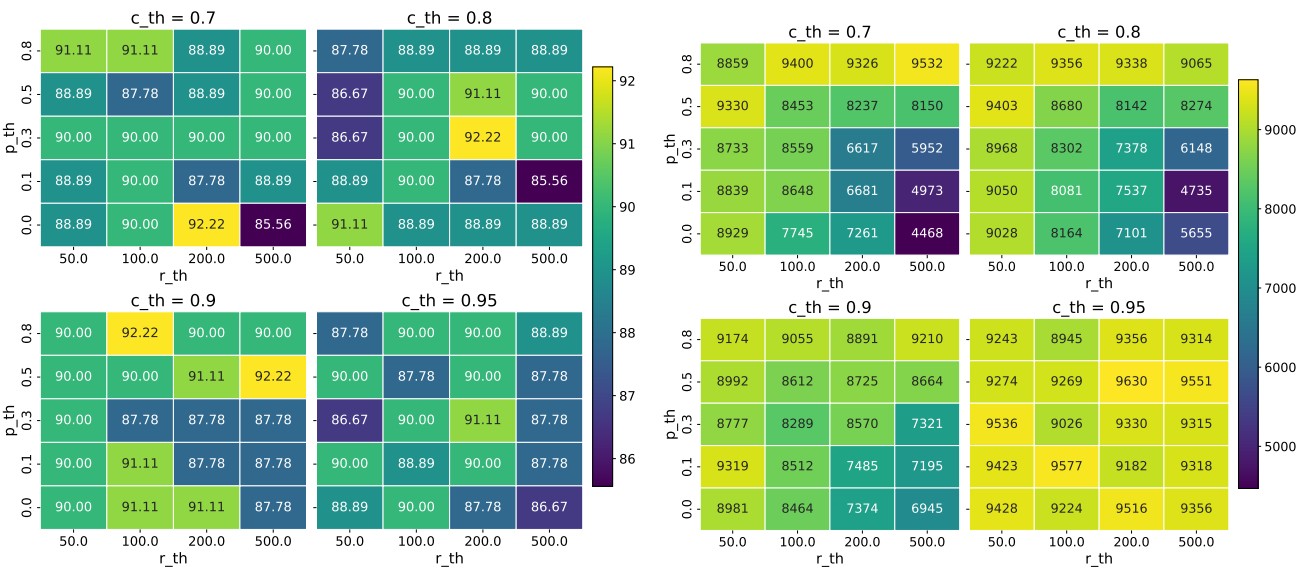

*(a)* Accuracy comparison.

*(b)* Output length comparison.

*Figure 9.* Accuracy and output length comparison in the calibration data under different threshold settings. The parameters c_th, p_th, and r_th denote the thresholds for confidence, progress, and remaining reasoning length estimation, respectively.

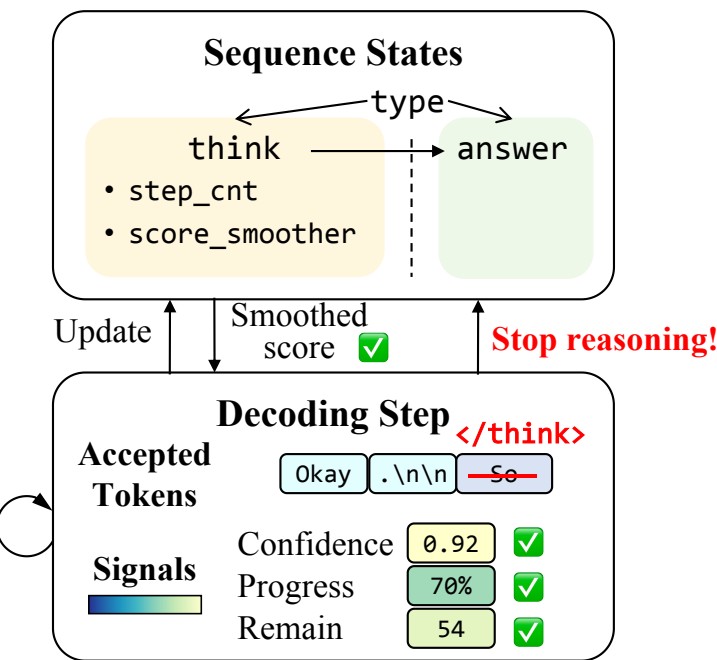

*Figure 10.* Inference process with signal-guided speculative exit.

**Signal Smoothing**: In the ablation study of signal smoothing strategies, the following methods are implemented to stabilize cognitive signals for early-exit decisions:

- Sliding Window: The sliding window approach smooths the signal by averaging the last $N$ predicted signal values, with $N$ set to 10. The mean score $x_t$ at decoding step $t$ is computed as:

$$x_t = \text{Mean}(s_t, N) = \frac{1}{N} \sum_{i=t-N+1}^{t} s_i, \tag{7}$$

where $s_i$ denotes the predicted signal value at decoding step $i$.

- Momentum-based Prediction: This method predicts the next score based on the momentum, which is calculated as the difference between $N$ consecutive signal values, with $N$ set to 10. The predicted score $x_t$ at decoding step $t$ is given by:

$$x_t = \text{Predict}(s_t, N) = s_{t-1} + \frac{1}{N-1} \sum_{i=t-N+1}^{t-1} (s_i - s_{i-1}). \tag{8}$$

- Paragraph Mean: In this approach, the score $x_t$ is calculated as the average of all predicted signal values within the current paragraph:

$$x_t = \frac{1}{T} \sum_{i=1}^{T} s_i, \tag{9}$$

where $T$ is the total number of steps in the current paragraph.

- Exponential Weighted Moving Average (EWMA): In this approach, the smoothing factor $\alpha$ is set to 0.1. The new score $x_t$ is updated based on the previous score $x_{t-1}$ and the current signal value $s_t$ as:

$$x_t = \text{EWMA}(s_t, x_{t-1}, \alpha) = \alpha \cdot s_t + (1 - \alpha) \cdot x_{t-1}. \tag{10}$$

**Discourse Markers**: We collect high-frequency words appearing at the beginning of model-generated sentences as discourse markers. As shown in Figure 11, we present examples of discourse markers extracted from Qwen3-4B-Thinking-2507 on the MATH dataset. Among them, transitional words such as "Wait" and "But" can be regarded as a subset of these high-frequency markers.

**Reproducibility Statement:** All implementation details, including training scripts, hyperparameter configurations, and inference scripts, are released in our open-source repository to facilitate reproducibility. Our code has been made publicly available in a GitHub repository at https://github.com/Tencent/AngelSlim.

**Training Cost Analysis:** The additional training cost introduced by SpecExit over standard EAGLE3 consists of two components: dataset construction and signal head training. We note that SpecExit operates within a single draft model (the EAGLE3 draft model), which itself requires training. The early-exit supervision is added on top of this existing speculative decoding training and can be trained jointly with the EAGLE3 objective at low marginal cost.

*Dataset Construction.* To determine optimal segmentation points for early-exit supervision, we evaluate each reasoning trace at up to five candidate paragraph-boundary positions, selected by evenly dividing all split indices. At each candidate position, we insert a `</think>` token, decode up to 500 tokens, and verify whether the predicted answer is correct. Each sequence therefore triggers at most five forward passes, generating up to 2,500 tokens in total. We measured this overhead using 5,000 randomly sampled examples from the MATH training set and executed the process through parallel vLLM service calls on an 8×H20 cluster. The complete data construction phase required 0.6 hours (4.8 H20 GPU-hours).

*Signal Head Training.* The auxiliary exit-signal heads add only a small linear projection applied to the hidden state. Let $x \in \mathbb{R}^{s \times d}$ denote the hidden state with sequence length $s$ and hidden dimension $d$, and let $W \in \mathbb{R}^{d \times 3}$ denote the linear projection weight. The output $y = xW$ has shape $(s, 3)$.

In the forward pass, the linear projection $y = xW$ costs $2 \times s \times d \times 3 = 6sd$ FLOPs, and the loss calculation involves element-wise operations (Sigmoid, MSE, MSLE) costing approximately $15s$ FLOPs. In the backward pass, the weight gradient $\partial \mathcal{L}/\partial W = x^\top (\partial \mathcal{L}/\partial y)$ costs $6sd$ FLOPs, and the input gradient $\partial \mathcal{L}/\partial x = (\partial \mathcal{L}/\partial y)W^\top$ costs $6sd$ FLOPs. Loss

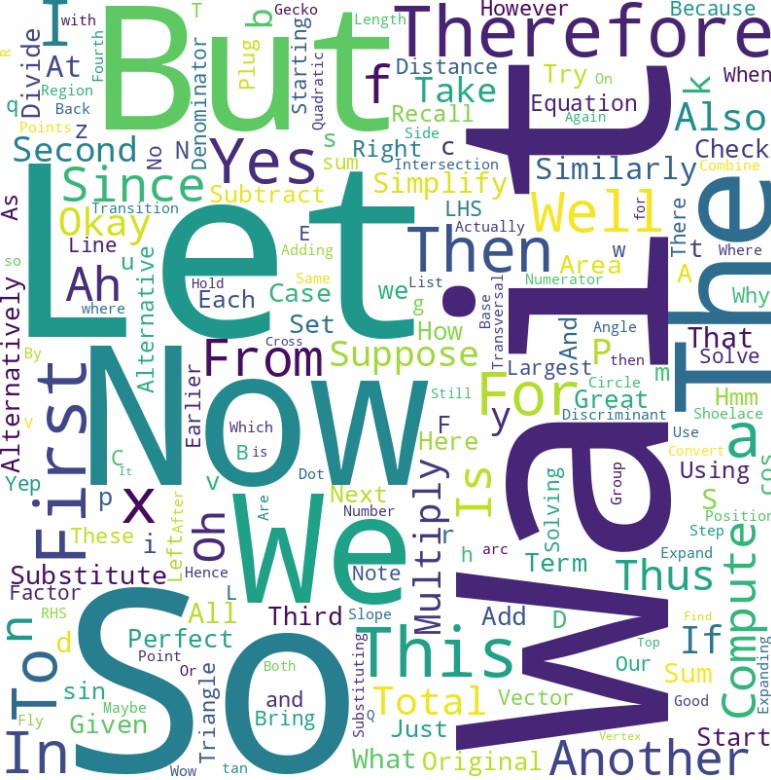

*Figure 11.* Discourse marker distribution in Qwen3-4B-Thinking-2507's responses on the MATH500 (Hendrycks et al., 2021) dataset.

gradient backpropagation adds $O(s)$ FLOPs. The total additional cost is therefore approximately $18sd + O(s)$ FLOPs per sequence, constituting a negligible fraction of the draft model forward pass.

Training a full epoch of the SpecExit modules (EAGLE3 combined with the early-exit module) requires approximately 1.5 hours (12 H20 GPU-hours). The exit-signal heads typically converge within 1 to 3 epochs. Because their regression objectives are orthogonal to the token-prediction objective, these heads can be trained jointly with the speculative decoding model or added through a short continued training phase.

Both sources of overhead are small compared to the runtime benefits. SpecExit achieves up to 66% reduction in reasoning length and $2.5\times$ end-to-end inference speedup over the EAGLE3 baseline, and the method can be integrated into existing speculative decoding training pipelines with minimal additional cost.

**Code Task Token Reduction:** Code generation tasks generally exhibit lower token reduction than mathematical reasoning tasks because code reasoning inherently contains less redundancy. Each reasoning step in code generation tends to contribute more directly to the solution structure, offering less room for early truncation. In our training data construction, the average vanilla reasoning length on MATH500 is 6934 tokens, which reduces to 2577 tokens after identifying the shortest valid path (62.8% reduction). In contrast, the average vanilla reasoning length on LeetCode-style tasks is 8915 tokens, which reduces to 5317 tokens (40.4% reduction). This difference reflects the lower prevalence of redundant verification and restated steps in code reasoning traces.

### A.3. Signal Prediction Error Analysis

To quantify the reliability of SpecExit's predicted signals, we compare predicted values against ground-truth signal values across datasets of varying difficulty. As shown in Figure 12, the predicted confidence tends to systematically overestimate the model's certainty on difficult tasks such as AIME, while predictions on easier datasets remain well-calibrated. This overestimation explains the conservative behavior of SpecExit on hard problems, where fewer early-exit opportunities are triggered.

The multi-signal fusion mechanism compensates for individual signal prediction errors. Because the early-exit decision requires all three signals (confidence, progress, and remaining length) to simultaneously satisfy their respective thresholds after EWMA smoothing, occasional overestimation in one signal is unlikely to cause premature termination unless corroborated by the other two. This redundancy provides robustness against individual signal inaccuracies and explains why the combined configuration (SpecExit*) maintains higher accuracy than any single-signal variant.

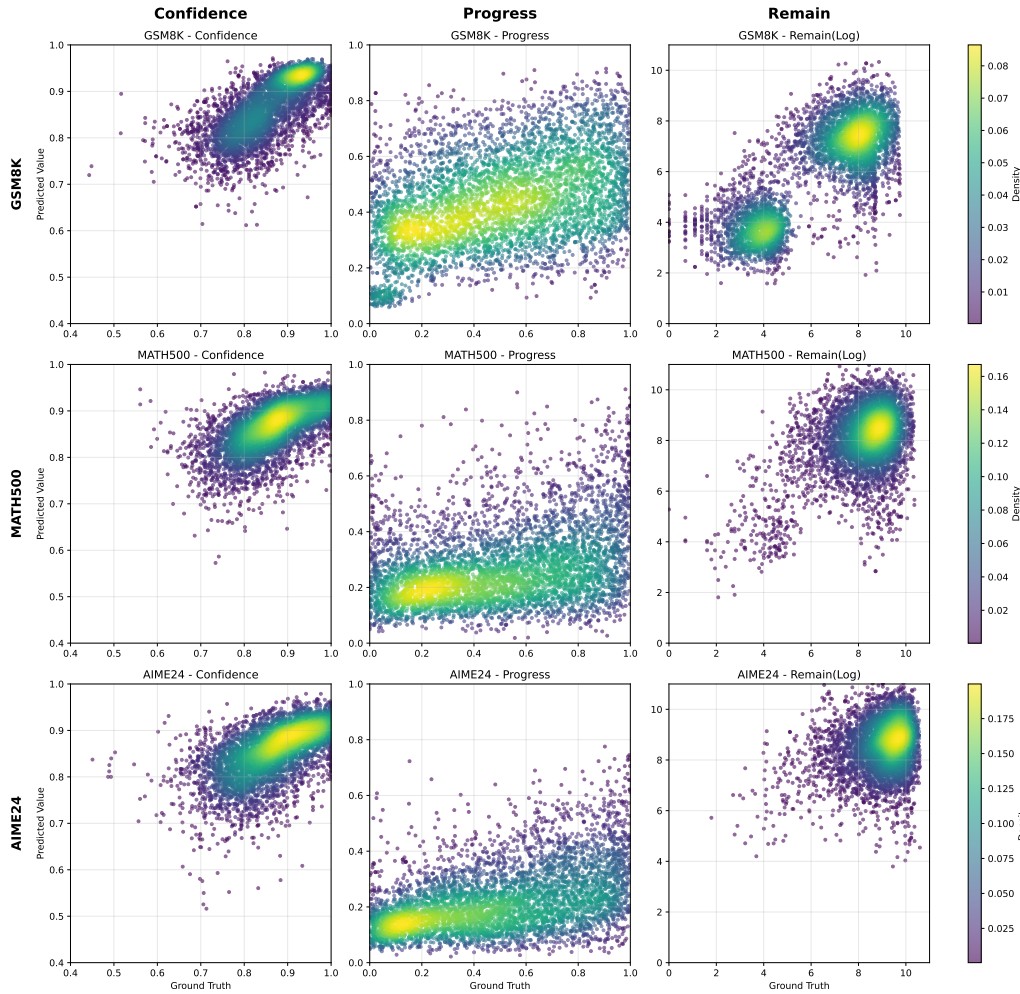

*Figure 12.* Comparison of predicted and ground-truth signal values across datasets of varying difficulty. The model tends to overestimate confidence on harder tasks (e.g., AIME), while predictions on easier tasks (e.g., GSM8K) are better calibrated.

## A.4. Predicted Signals Visualization

As shown in Figure 13, for simple problems, the predicted confidence remains high, the predicted remaining reasoning length is relatively short, and the predicted progress rises rapidly within the first few sentences, with only minor drops on a few uncertain words. This indicates that for such problems, the model is able to establish a stable reasoning trace at an early stage. In contrast, as shown in Figure 14, for complicated problems, the model also exhibits high confidence and a short remaining reasoning length in the initial summarization phase, but once it enters the detailed analysis stage, the predicted remaining reasoning length increases significantly, confidence drops. Meanwhile, the predicted progress starts low and rises initially, but then fluctuates markedly during repeated self-reflection, making it difficult to stabilize at a high threshold.

These observations reveal the inherent limitations of relying on individual signals for early exiting. When depending solely on confidence, the model often exhibits overconfidence and terminates too early before sufficient reasoning has been completed, leading to substantial accuracy degradation. When depending solely on the predicted remaining reasoning length,

the model may become overly optimistic in the early stages of complicated problems, resulting in premature exits before essential reasoning steps are accomplished. When depending solely on progress, the signal tends to fluctuate and remain unstable in complex reasoning tasks, making it difficult to trigger an appropriate early-exit and thereby restricting achievable speedup. In summary, each single signal suffers from the inability to balance accuracy and efficiency across diverse problem types. By integrating multiple signals in a complementary manner, the model can achieve a smoother trade-off between reasoning accuracy and inference acceleration.

Figure 15 illustrates the effect of applying an Exponentially Weighted Moving Average (EWMA) to stabilize predicted signal. EWMA effectively suppresses local noise while preserving the global trend, leading to smoother traces for signal. The smoothed signals reveal clearer overall convergence patterns, providing a more stable and reliable basis for threshold-based early-exit decisions and thereby improving robustness in challenging reasoning scenarios.

### A.5. Case Study Details

Figure 16 presents an example from the GSM8K (Cobbe et al., 2021) dataset, where SpecExit is applied after an initial analysis. In this case, SpecExit inserts a decision to exit reasoning based on the signal magnitude after completing the first paragraph, thus preventing the continuation of redundant reasoning tokens. The process begins with the model evaluating the initial segment of the problem, analyzing the available context and producing intermediate reasoning steps. When SpecExit detects that all the signals exceeds a predefined threshold, it triggers the early-exit mechanism. As a result, the model halts further reasoning and transitions directly to the answering phase, thus producing the correct result in a more efficient manner.

By shortening the unnecessary token generation and reducing redundant reasoning, SpecExit leads to significant performance improvements in terms of inference speed while maintaining the correctness of the answer. This early-exit strategy not only accelerates the inference process but also avoids the potential confusion that the model might experience from excessive reflection or unnecessary computations. This case highlights the effectiveness of integrating early-exit mechanisms in enhancing both the efficiency and reliability of large language model inference.

**Failure Case:** Figure 17 presents a failure case from MATH500 (intermediate_algebra-1555), where SpecExit triggers an early exit prematurely. In this example, the model terminates reasoning before completing a self-verification step that would have caught an arithmetic error. The predicted confidence and progress signals both exceed their thresholds at a paragraph boundary, but the subsequent reasoning (which was truncated) contained a necessary error-correction step. This case illustrates that when the model's reasoning involves iterative self-verification, premature termination can remove the opportunity for error detection and correction.

**Uncertainty Increase During Reasoning:** Figure 18 shows a case from MATH500 (intermediate_algebra-607) where the predicted signals reflect increasing uncertainty during the reasoning process. As the model encounters a challenging sub-problem, the confidence signal decreases and the remaining length estimate increases, correctly indicating that the reasoning is far from complete. In this case, SpecExit appropriately refrains from triggering an early exit, allowing the model to continue reasoning until sufficient evidence is accumulated. This example demonstrates that the predicted signals can capture transitions in reasoning difficulty and adaptively delay the exit decision.

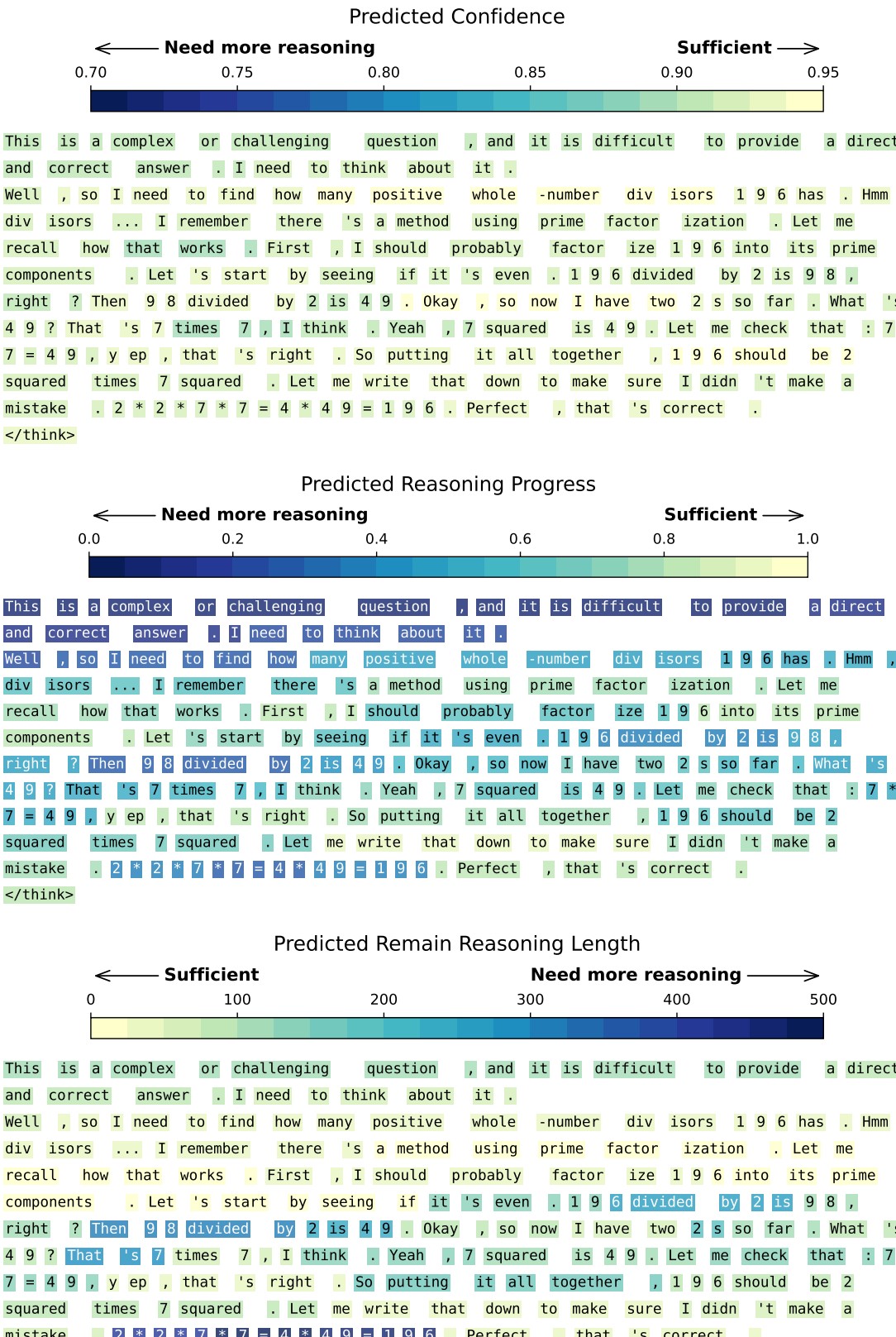

*Figure 13.* Visualization of reasoning signals for a simple problem, illustrated with an example from the MATH500 (Hendrycks et al., 2021) dataset, where darker colors denote insufficient reasoning and lighter colors denote sufficiency.

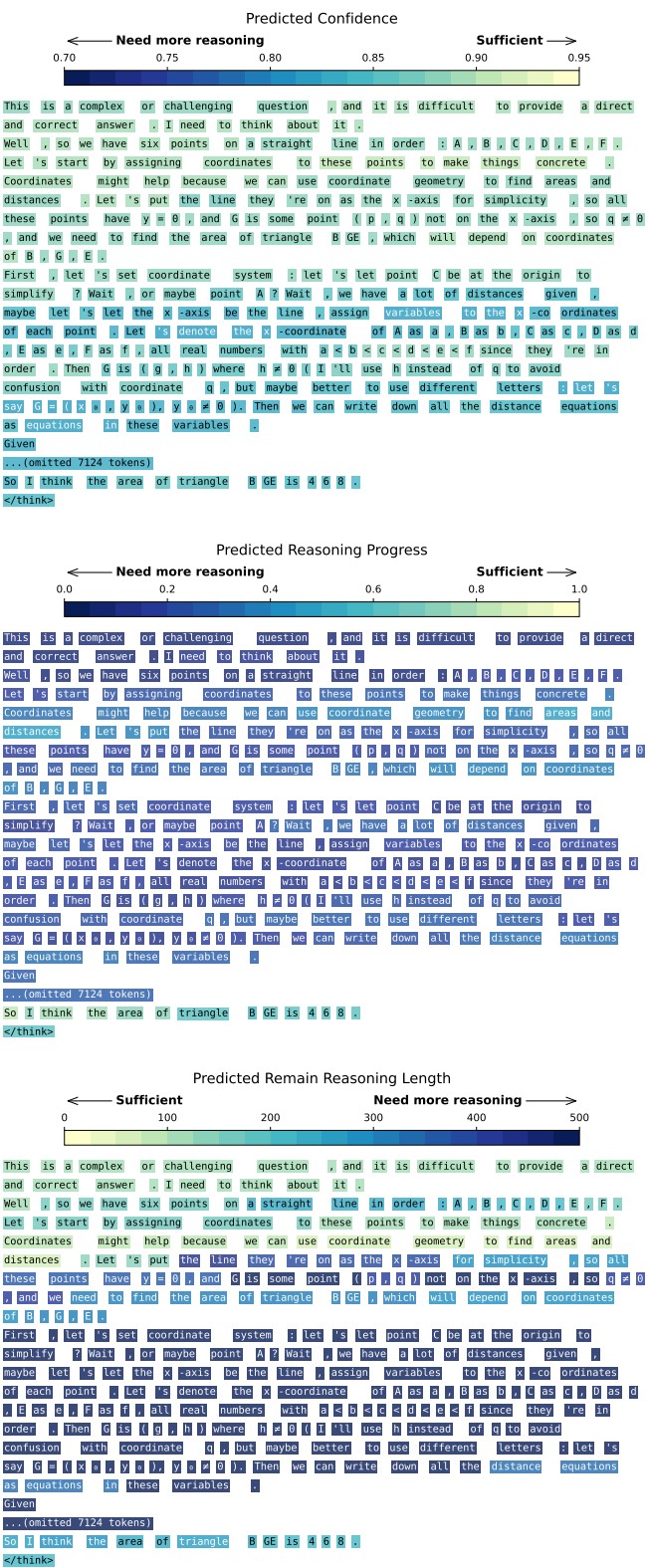

*Figure 14.* Visualization of reasoning signals for a complicated problem, illustrated with an example from the AIME (MAA Committees) dataset, where darker colors denote insufficient reasoning and lighter colors denote sufficiency.

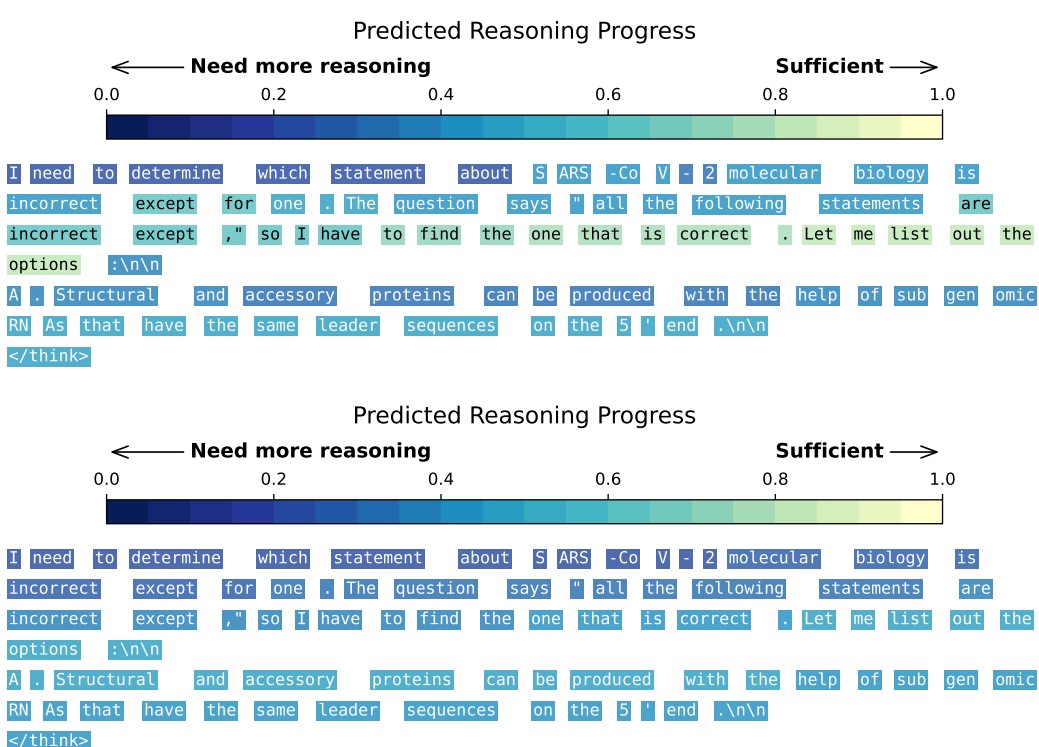

*Figure 15.* Visualization of the smoothing effect introduced by the EWMA-based signal estimator, illustrated with an example from the GPQA Diamond (Rein et al., 2024) dataset, where the stabilized trace reveals reduced high-frequency fluctuations and clearer monotonic trends in the underlying reasoning signals.

## Question:

Katerina makes designer jewelry. Her specialty is topaz necklaces. She uses 8 topaz gemstones per necklace, and fills the space between gemstones using sterling silver beads. If each topaz gemstone is one inch long, each sterling silver bead is one-quarter of an inch long, and each necklace is made to a total length of 25 inches, how many sterling silver beads does Katerina use per necklace?

## SpecExit: **(421 tokens)**

Okay, let's try to figure out how many sterling silver beads Katerina uses per necklace. Hmm, first, let's list out what we know.\n\nShe uses 8 topaz gemstones per necklace. Each topaz is 1 inch long, so the total length of the topaz gemstones would be 8 times 1 inch, right? So that's 8 inches.\n\n
</think>     **Signals: confidence=0.94 progress=41% remain_tokens=56** ✅

To determine how many sterling silver beads Katerina uses per necklace... (omitted)
### Final Answer\n\n$$\n\\boxed{68}\n$$\n\n
Katerina uses **68** sterling silver beads per necklace.     **correct** ✅

## Vanilla: **(1521 tokens)**     **Initial**

Okay, let's try to figure out how many sterling silver beads Katerina uses per necklace. Hmm, first, let's list out what we know.\n\nShe uses 8 topaz gemstones per necklace. Each topaz is 1 inch long, so the total length of the topaz gemstones would be 8 times 1 inch, right? So that's 8 inches.

**Analyzation**

Hmm, let's start by breaking down the problem step by step. \n\n
First, the problem says she uses 8 topaz gemstones per necklace, and each topaz is one inch long. So, the total length of the topaz gemstones would be 8 times 1 inch, right? Let me write that down: 8 topazes * 1 inch each = 8 inches.\n\n
Now, the necklace is supposed to be a total length of 25 inches. That means the space between the gemstones, which is filled with sterling silver beads, has to make up the rest of the length. So, the total length from the silver beads would be 25 inches minus the length of the topaz gemstones. Let me calculate that: 25 inches - 8 inches = 17 inches. So, the silver beads together are 17 inches long.

**Reflection**

But each sterling silver bead is one-quarter of an inch long... (omitted)
Wait, hold on. When you have beads between gemstones, do you need to consider how many gaps there are? Because ... (omitted)
Oh! That's a crucial point I might have missed. The problem says \"fills the space between gemstones using sterling silver beads.\" So, the silver beads are in the spaces between the topaz stones. So, if there are 8 topaz stones, there are 7 spaces between them.\n\n
Let me re-examine the problem statement...(omitted)
Wait, but the problem says \"each necklace is made to a total length of 25 inches\". So, total length = length of topaz + length of silver beads.\n\n
But ...(omitted) (continuous reflection over 26 paragraphs)

**Summary**

So, in the original problem, 8 topaz stones (8 inches), total necklace 25 inches, so 17 inches of silver beads. 17 / 0.25 = 68.\n\n
I think that's the answer. The key is that the problem says \"fills the space between gemstones using sterling silver beads\", so the total length of the beads is the total necklace length minus the topaz length, and then divide by the bead length.\n\n
I think the confusion might be about the number of gaps, but since the problem doesn't specify that each gap has a certain number of beads, we just take the total length of the beads and divide by the length per bead.\n\n
So, the answer should be 68.
</think>

*Figure 16.* A case study from the GSM8K dataset (Cobbe et al., 2021), comparing SpecExit and vanilla generation.

## Predicted Reasoning Progress

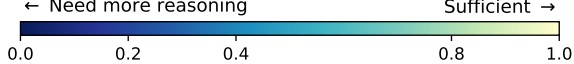

← Need more reasoning                  Sufficient →

0.0        0.2        0.4                0.8        1.0

```
Question: Find the remainder when $(5x + 9)^{611} + (x + 5)^{11} + (x - 1)^{11}
+ 3x^2 + 1$ is divided by $x + 2.$
```

```
L00   This is a complex or challenging  question , and it is difficult  to provide  a
L01   direct  and correct  answer . I need  to  think  about  it .\n
L02   Well , so I need  to  find  the  remainder  when  this  big  polynomial   is  divided  by  x +
L03   2. Hmm , when  dividing  by a linear  polynomial   like  x + a, I remember  there 's the
L04   Remain der  The orem , right ? Wait , let  me  recall  :  the  Remain der  The orem  says  that
L05   the  remainder  when  a polynomial   f(x) is  divided  by  (x - c) is  f(c). But  here  it
L06   's x + 2, which  is  x - (- 2), so  that  means  the  remainder   should  be  f(-2), where
L07   f(x) is  the  polynomial   given . Is  that  right ? Let  me  make  sure  I didn 't mix  up
L08   the  signs .\n\n
L09   </think>  \n\n
L10   To find  the  remainder   when  a polynomial   $ f(x) $ is  divided  by $ x + 2 $, we can
L11    apply  the  ** Rem ain der  The orem **. This  theorem  states  that :\n\n
L12   > If a polynomial   $ f(x) $ is  divided  by $ x - c $, the  remainder   is $ f(c) $
L13   .\n\n
L14   In this  case , the  divisor  is $ x + 2 $, which  is  equivalent  to $ x - (- 2) $.
L15    Therefore  , the  remainder  when  $ f(x) $ is  divided  by $ x + 2 $ is  simply  $ f(-2
L16   ) $.\n\n
L17   --- \n\n
L18   Let 's define  the  polynomial   as :\n
L19   $$ \n
L20   f(x) = (5x + 9)^{611} + (x + 5)^{11} + (x - 1)^{11} + 3x^2 + 1\n
L21   $$ \n\n
L22   We now  compute   $ f(-2) $ step  by step .\n\n
L23   --- \n\n
L24   ### Step  1: Evaluate   each  term  at $ x = -2 $\n\n
L25   #### Term  1: $ (5x + 9)^{611} $\n
L26   $$ \n
L27   5(-2) + 9 = -10 + 9 = -1 \Rightarrow   (- 1)^{611} = -1\n
L28   $$ \n\n
L29   #### Term  2: $ (x + 5)^{11} $\n
L30   $$ \n
L31   (- 2) + 5 = 3 \Rightarrow   3^{11}\n
L32   $$ \n\n
L33   #### Term  3: $ (x - 1)^{11} $\n
L34   $$ \n
L35   (- 2) - 1 = -3 \Rightarrow   (- 3)^{11} = -3^{11}\n
L36   $$ \n\n
L37   #### Term  4: $ 3x^2 $\n
L38   $$ \n
L39   3(-2)^2 = 3(4) = 12\n
L40   $$ \n\n
L41   #### Term  5: Constant   term \n
L42   $$ \n
L43   1\n
L44   $$ \n\n
L45   --- \n\n
L46   ### Step  2: Combine   all  terms \n\n
L47   Now plug  in the  values  :\n\n
L48   $$ \n
L49   f(-2) = (-1) + 3^{11} + (- 3^{11}) + 12 + 1\n
L50   $$ \n\n
L51   Simplify   term  by term :\n\n
L52   - $ -1 + 1 = 0 $\n
L53   - $ 3^{11} - 3^{11} = 0 $\n
L54   - $ 12 + 1 = 13 $\n\n
L55   So :\n
L56   $$ \n
L57   f(-2) = 0 + 0 + 13 = 13\n
L58   $$ \n\n
L59   --- \n\n
L60   ### Final  Answer \n\n
L61   $$ \n
L62   \boxed {13}\n
L63   $$
```

*Figure 17.* A failure case from MATH500 (intermediate_algebra-1555). SpecExit exits prematurely before a self-verification step, causing an arithmetic error in the final answer.

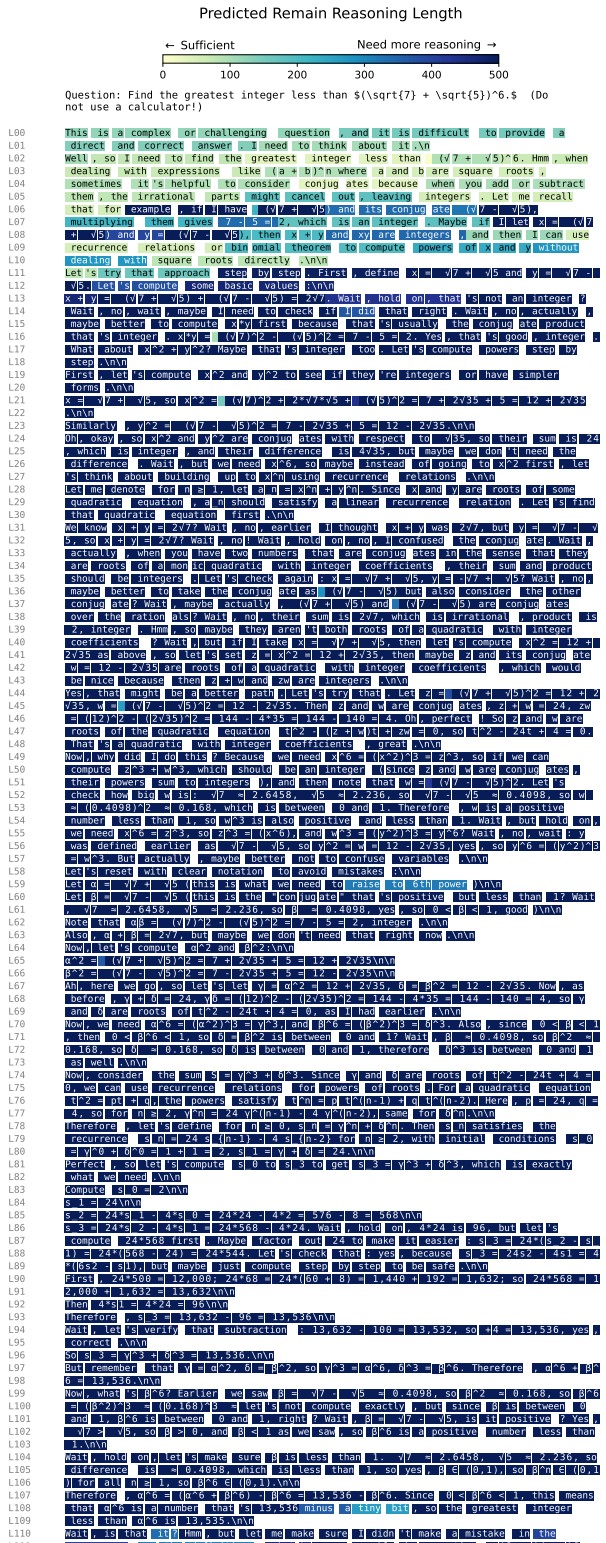

*Figure 18.* A case from MATH500 (intermediate_algebra-607) where predicted signals correctly reflect increasing uncertainty, preventing premature exit.

