# OpenReview forum: "SpecExit: Accelerating Large Reasoning Model via Speculative Exit"
_ICML.cc/2026/Conference — ICML 2026 regular_

### Official Review · Reviewer_BG5w · 2026-03-08

**Soundness:** 2
**Presentation:** 3
**Significance:** 2
**Originality:** 2
**Overall Recommendation:** 4
**Confidence:** 4

**Summary:**

This paper proposes SpecExit, a framework that combines speculative decoding with reasoning-aware early exit for inference acceleration. The core approach attaches auxiliary prediction heads to the draft model's MTP layer, enabling simultaneous output of token distributions and three reasoning signals (confidence, reasoning progress, and remaining reasoning length) in a single forward pass, without modifying the target model. During training, multi-task learning jointly optimizes token classification loss and signal regression loss. At inference, EWMA-smoothed signals trigger early exit at paragraph delimiters. Experiments on GSM8K, MATH500, AIME, HumanEval+, GPQA-D, and ARC-Challenge across three LRMs report up to 66% generation length reduction and 2.5× end-to-end speedup.

**Compliance With Llm Reviewing Policy:**

Affirmed.

**Key Questions For Authors:**

**Q1 (Section 3.2 and Figure 5) :** Can you provide quantitative error metrics (MAE, Pearson correlation) between predicted signal values and ground-truth labels? Figure 5 shows severe accuracy degradation for confidence-only on AIME—what is the compensation mechanism when multiple signals are fused?

**Q2 (Table 1):** For DeepSeek-R1-Distill-Llama-8B on GSM8K, how do you explain that EAGLE3 baseline (79.3%) outperforms Vanilla (76.4%)? Does SpecExit's lower accuracy (75.3%) compared to EAGLE3 indicate negative coupling between early exit and speculative decoding precision?

**Q3  (Section 3.2 and HumanEval+ results):** How is "shortest valid reasoning path" defined for HumanEval+? Does inserting `</think>` alter the final code output, and what are the correctness verification criteria?

**Q4  (Section 4.1 and comparison with EAGLE3):**  Compared to using an off-the-shelf EAGLE3 draft model, what are the specific costs of SpecExit's joint training (training data scale, signal annotation method, training duration)?

**Q5 (Section 4.1, Table 1, and Appendix A.1):** SpecExit uses unified thresholds across all benchmarks, but different tasks have vastly different reasoning length distributions. Can you provide comparative experiments with task-specific threshold calibration to assess the performance gap between unified and task-adaptive thresholds?

**Limitations:**

The authors' discussion of limitations is insufficient. The following should be explicitly addressed:

- Draft model MTL retraining costs
- Cross-task transferability of threshold calibration and additional overhead for deployment on new tasks
- Rationale and limitations of label construction logic for code generation tasks like HumanEval+
- Impact of signal prediction error on early-exit decision reliability, particularly on out-of-distribution tasks

**Strengths And Weaknesses:**

**Strengths**

- The paper directly demonstrates through Figure 1a that DEER exhibits higher end-to-end latency than Vanilla on GSM8K and ARC, accurately identifying the core deficiency of probing-based methods: probing overhead negates the benefits of length reduction. The experimental evidence is direct and compelling.
- The experimental setup covers three mainstream LRM families and six heterogeneous benchmarks. Ablation studies systematically examine signal types, smoothing strategies, and delimiter selection. The experiments are comprehensive with a clear logical chain.

**Weaknesses**

- The paper only qualitatively demonstrates through PCA/UMAP visualization that hidden states contain reasoning signals, but does not quantify the error between predicted signal values and ground-truth labels. Figure 5 shows that the confidence-only variant experiences significant accuracy degradation on AIME (76.7% vs 86.7%), indirectly indicating systematic bias in this signal on complex tasks, yet the paper provides no quantitative explanation.

- Section 3.2 defines "shortest valid reasoning path" as the shortest reasoning segment where inserting `</think>` maintains consistency with the original output. While understandable for math tasks, this definition is questionable for code generation. The paper provides almost no specific explanation for label construction on HumanEval+, yet SpecExit's token reduction on this benchmark (Qwen3: 15%, Phi-4: 22%) is notably lower than on math tasks. This phenomenon may reflect label quality issues but remains undiscussed.

- Section 4.1 states that the EAGLE3 baseline's draft model uses the same training procedure as SpecExit's draft model, implying SpecExit incurs additional MTL training costs (data volume, training time) compared to the original EAGLE3, yet this is not discussed in the main text.

  The DSE calibration in Appendix A.1 requires approximately 2.5 hours on 8×H20 and must be run for specific models and tasks. In current experiments, all benchmarks share the same thresholds, but different tasks have significantly different difficulty distributions—the rationale for unified thresholds is not discussed. The paper provides no cross-task threshold transfer experiments and lacks evaluation of deployment costs on new tasks.

---

> ### Author Rebuttal · Authors · 2026-03-30
>
> We thank the reviewer for the insightful and valuable comments. We respond to each point below.
>
>
> - **Q1: Prediction error analysis**
>
>     Thank you for pointing out the lack of this analysis. We have conducted a quantitative comparison between predicted and ground truth signal values across datasets of varying difficulty, as shown in this [Figure](https://anonymous.4open.science/r/SpecExit-B802/figs/error_analysis.pdf). On the more challenging AIME dataset, the predicted confidence values tend to be systematically higher than the ground truth, while the progress signal evolves more slowly. This is consistent with your observation that the confidence-only variant suffers from significant accuracy degradation on AIME. We will incorporate this analysis into the revised version of the manuscript.
>
>     Regarding the compensation mechanism, SpecExit applies EWMA smoothing and requires all three signals to simultaneously satisfy their respective thresholds before triggering early exit. Detailed case level analyses of signal trajectories for problems with different difficulty levels are provided in Figures 12–13 in Appendix A.3.
>
>
> - **Q2: Accuracy fluctuation**
>
>    Thank you for this careful observation. We have conducted additional reproductions for this accuracy fluctuation case. The results reported in Table 1 are based on three runs. Below are the 16-run reproduction results (DeepSeek-R1-Distill-Llama-8B, GSM8K, temperature 0.6). Notably, SpecExit achieves a slightly higher acceptance length (4.47->4.63), indicating that the early-exit mechanism does not degrade speculative decoding quality. In summary, the accuracy fluctuation observed in Table 1 can be attributed to statistical variation arising from evaluation randomness. Across the 3 models and 6 benchmarks we experimented on, SpecExit has accuracy no less than EAGLE3 for 10 of the 18 settings, confirming that there is no systematic negative coupling effect.
>
>     | Method | Paper (Table 1) | Reproduced (N=16) | Std | Accept Length | Speedup vs Vanilla |
>     |:--|:--:|:--:|:--:|:--:|:--:|
>     | Vanilla | 76.4% | 77.4% | ±0.9% | — | 1.00× |
>     | EAGLE3 | 79.3% | 76.6% | ±0.8% | 4.47 | 3.13× |
>     | SpecExit (Ours) | 75.3% | 74.6% | ±0.9% | 4.63 | 5.18× |
>
>
> - **Q3: Label construction for HumanEval+**
>
>     For training data construction on code tasks, the "shortest valid reasoning path" is defined analogously to math tasks: we insert `</think>` at candidate paragraph boundaries and verify correctness by executing the generated code against the full test suite. Only positions where all test cases pass are considered valid, making the criterion deterministic and objective.
>
>     Regarding the lower token reduction on code tasks, we appreciate your insightful observation. During training data construction, we find that code reasoning inherently offers less room for length reduction. The table below shows average token counts on two open-source training datasets:
>
>     | | MATH500 | LeetCodeDataset |
>     |--|--|--|
>     | Vanilla | 6,933.9 | 8,915.3 |
>     | Training Data| 2,577.0 | 5,316.6 |
>
>
> - **Q4: Training cost**
>
>     **Dataset construction**: We evaluate each reasoning trace at up to five candidate paragraph boundary positions, insert `</think>`, decode up to 500 tokens, and verify correctness. Using 5,000 MATH examples on 8$\times$H20, this phase required **0.6 h (4.8 H20 GPU hours)**.
>
>     **Signal training overhead**: The auxiliary exit-signal heads add only a linear projection $W \in \mathbb{R}^{d \times 3}$ applied to the hidden state $x \in \mathbb{R}^{s \times d}$. The forward pass computes $y = xW$ at $6sd$ FLOPs plus $\approx 15s$ FLOPs for loss computation. The backward pass computes $dL/dW = x^T(dL/dy)$ at $6sd$ FLOPs and $dL/dx = (dL/dy)W^T$ at $6sd$ FLOPs. The total additional cost is therefore $\approx 18sd + O(s)$ FLOPs per sequence, constituting a negligible fraction of the draft model's forward pass.
>
> - **Q5: Cross-domain threshold transferability**
>
>     All Table 1 results already use unified thresholds across all six benchmarks per model. This is a deliberate design choice motivated by practical deployment considerations: in real-world serving, the domain and difficulty of incoming queries are unknown in advance, making task specific threshold calibration impractical. SpecExit maintains comparable accuracy across heterogeneous benchmarks under a single threshold set, demonstrating cross domain robustness.
>
>     Regarding deployment costs on new tasks or models, exit-signal heads can be added to an existing EAGLE3 draft model through short continued training with convergence typically within 1 to 3 epochs.
>
> - **Limitations**
>
>     Thank you for the detailed suggestions. We will address the following in the revised manuscript: (1) additional MTL training cost, (2) rationale and scope of unified threshold calibration, (3) label construction characteristics for code generation tasks, and (4) signal prediction error analysis.

---

> > ### Author Rebuttal · Reviewer_BG5w · 2026-04-01
> >
> > Thank you for your rebuttal. I have no more questions and I'd like to maintain my score.

---

### Official Review · Reviewer_oHYv · 2026-03-09

**Soundness:** 3
**Presentation:** 3
**Significance:** 3
**Originality:** 3
**Overall Recommendation:** 4
**Confidence:** 3

**Summary:**

This paper proposes SpecExit, a framework that addresses the overthinking problem in large reasoning models by extending the draft model's multi-token prediction layer with auxiliary heads that predict confidence, reasoning progress, and remaining reasoning length from hidden states. This enables early termination of chain-of-thought generation within the speculative decoding pipeline without probing overhead or target model modification. Evaluated on three model families across six benchmarks, SpecExit achieves up to 66% token reduction and 2.5x end-to-end speedup while maintaining comparable accuracy.

**Compliance With Llm Reviewing Policy:**

Affirmed.

**Key Questions For Authors:**

1. Have you measured how much accuracy degrades when thresholds calibrated on one dataset (e.g., GSM8K) are applied to a different domain (e.g., GPQA or HumanEval)? Cross-domain transferability is essential for practical deployment.

2. Is there a minimum reasoning length constraint before the exit logic activates?

3. How does SpecExit interact with batched/continuous-batching inference in vLLM? Early-exiting one sequence in a batch may leave GPU resources underutilized — have you measured throughput in multi-request settings?

**Limitations:**

yes

**Strengths And Weaknesses:**

### Strengths

1. **Well-motivated and clean design.** The paper clearly identifies that probing-based early-exit methods fail to deliver latency gains despite reducing token count . Piggy-backing exit signals onto the existing draft model's MTP layer adds negligible overhead and requires no target model changes — an elegant solution.

2. **Strong empirical validation.** Experiments span comprehensive benchmarks, and report latency. The ablation studies are thorough and provide actionable insights.

---

### Weaknesses

1. **Training data construction has limited scalability.** Iteratively inserting `</think>` after each paragraph and verifying answer correctness only works for tasks with verifiable outputs, limiting applicability to open-ended generation or dialogue tasks.

2. **Missing failure mode analysis.** The paper does not examine cases where SpecExit exits prematurely and produces wrong answers. Understanding which problem types or reasoning patterns cause failure would significantly strengthen the contribution. No worries on the bad cases, it always happens. But categorizing them into some typical classes can help future works.

---

> ### Author Rebuttal · Authors · 2026-03-30
>
> We thank the reviewer for the insightful and valuable comments. We respond to each point below.
>
> - **Weakness 1: Scalability of training data construction**
>
>     Thank you for this insightful observation. We acknowledge that our current label construction relies on verifiable outputs and is therefore best suited for tasks with deterministic correctness criteria. For simple question-answering tasks, we adopt temperature=0 decoding and verify correctness by comparing answer similarity. Extending SpecExit to open-ended generation tasks would require alternative supervision strategies, such as reward model scores or semantic similarity. We consider this a potentially important direction for future work and will discuss this limitation in the revised paper.
>
>
> - **Weakness 2: Failure case analysis**
>
>     Thank you for this constructive suggestion. We have conducted a failure case analysis, as shown in this [Figure](https://anonymous.4open.science/r/SpecExit-B802/case_study/math500-intermediate_algebra-1555/specexit-signal_progress.pdf). In this case, SpecExit determines that reasoning is sufficient and triggers early-exit. However, during the subsequent answer generation, the model makes a minor arithmetic error where a term "+1" is counted twice, and because the reasoning phase has already terminated, the model has no opportunity to self-correct through further reflection.
>
>     This case illustrates that premature early exit may deprive the model of the self-verification steps needed to catch minor computational errors. More generally, there exists an inherent trade-off between reducing reasoning length and maintaining answer correctness, which our threshold mechanism is designed to balance. As a potential direction for future work, a brief reflective verification phase could be introduced after the exit decision and before final answer generation.
>
> - **Q1: Cross-domain threshold transferability**
>
>     All results in Table 1 already use unified thresholds across all six benchmarks for each model. This is a deliberate design choice motivated by practical deployment considerations: in real-world serving, the domain and difficulty of incoming queries are unknown in advance, making task specific threshold calibration impractical. The fact that SpecExit maintains comparable accuracy across heterogeneous benchmarks under a single set of thresholds demonstrates its cross domain robustness. We will make this design rationale more explicit in the revised manuscript.
>
> - **Q2: Minimum reasoning length**
>
>     We do not impose any explicit constraint on the minimum number of tokens. Early exit decisions are made only at paragraph boundaries, so the minimum reasoning length corresponds to one complete paragraph.
>
> - **Q3: Interaction with continuous batching in vLLM**
>
>     SpecExit is integrated into the EAGLE speculative decoding implementation in vLLM. All main results reported in Table 1 are measured under continuous batching and multi-request settings. When a sequence exits early, vLLM’s continuous batching scheduler automatically fills the freed slot with pending requests, ensuring that GPU utilization is not degraded. In fact, early exit can further improve overall throughput by releasing resources for queued requests sooner.

---

> > ### Author Rebuttal · Reviewer_oHYv · 2026-04-03
> >
> > Thanks for the reply! I’ll maintain the already positive score.

---

### Official Review · Reviewer_zyU4 · 2026-03-21

**Soundness:** 3
**Presentation:** 3
**Significance:** 3
**Originality:** 2
**Overall Recommendation:** 4
**Confidence:** 3

**Summary:**

This work proposes SpecExit, a novel method that introduces lightweight adapters to reduce the inference overhead of LRMs.

**Compliance With Llm Reviewing Policy:**

Affirmed.

**Final Justification:**

The authors’ response did not resolve all of my concerns, but I decide to maintain my positive score.

**Key Questions For Authors:**

1.Why are training-based approaches not included in Table 1 for comparison? I believe the reported “66% generation length reduction and 2.5× end-to-end speedup” would be more convincing if the comparison included the state-of-the-art methods across the entire subfield, rather than only the speculative decoding baseline.

2.It is unclear to me whether the motivational results in Figure 1b sufficiently support the stated conclusion. In particular, it would be helpful to investigate whether uncertainty might increase during the reasoning process. Could the authors provide more detailed visualizations, for example, of a redundant reasoning process, to better illustrate this behavior?

**Limitations:**

The paper does not discuss the limitations of the proposed method. It would be helpful if the authors could briefly address potential limitations or scenarios where SpecExit might be less effective.

**Strengths And Weaknesses:**

Strengths:

1.The paper is easy to follow.

2.The work is well-motivated, particularly in Figure 1a, which provides a fine-grained analysis of the latency overhead issue in probing-based early-exit methods.

Weaknesses:

1.Some parts of the writing could be clarified. For example, the reasoning progress metric should be briefly explained or referenced near Figure 1b to help the reader understand how it is measured.

2.The experiments could be more comprehensive to better demonstrate the effectiveness of the proposed method.

---

> ### Author Rebuttal · Authors · 2026-03-30
>
> We thank the reviewer for the insightful and valuable comments. We respond to each point below.
>
> - **Q1: Comparison with training-based approaches**
>
>     SpecExit is designed for scenarios where practitioners aim to accelerate inference **without modifying the target model's parameters**. Only a lightweight draft model is trained at low cost to provide both speculative decoding acceleration and early-exit capability.
>     In contrast, training-based approaches modify the base model’s reasoning behavior through fine-tuning or reinforcement learning, reflecting a fundamentally different design goal and deployment setting. Moreover, such approaches may risk destabilizing the model’s output distribution, potentially degrading generation quality.
>     Therefore, the two categories of methods are complementary rather than directly comparable. SpecExit reduces redundant computation at low cost, as it does not require training the target model, and preserves the original model’s full reasoning capability. In contrast, training-based methods alter the model’s intrinsic reasoning patterns.
>
> - **Q2: Figure 1b and detailed visualizations**
>
>     Thank you for this valuable suggestion. Figure 1b is intended as a concise illustration showing that reasoning relevant signals can be learned from hidden states. Specifically, the reasoning progress metric denotes the normalized position of each reasoning token within the entire reasoning trace. The definition and implementation details are provided in Section 3.2. Additional analyses are provided in the Appendix, including signal values across problems of varying difficulty and case studies demonstrating how our method reduces redundant reasoning. In the revised manuscript, we will refine the explanation near Figure 1b in the main text and include an explicit reference to Section 3.2 for clarity.
>
>     We have additionally prepared case studies to further illustrate this behavior. As shown in this [Figure](https://anonymous.4open.science/r/SpecExit-B802/case_study/math500-intermediate_algebra-623/specexit-signal_remain.pdf), SpecExit terminates reasoning after the model obtains the key information $\log(x^2)$.  Notably, the **darker** background indicates **lower predicted reasoning sufficiency**. In contrast, the vanilla output shown in this [Figure](https://anonymous.4open.science/r/SpecExit-B802/case_study/math500-intermediate_algebra-623/origin-response.pdf) continues with substantial **redundant reflection** , supporting our motivation. For a more complicated question, as shown in this [Figure](https://anonymous.4open.science/r/SpecExit-B802/case_study/math500-intermediate_algebra-607/specexit-signal_remain.pdf), the signals indicate that the model is relatively certain during the initial problem summarization stage, while this certainty quickly decreases as the model enters more uncertain reflection stages. This aligns with the "uncertainty increase" phenomenon you mentioned. We will incorporate these detailed case analyses into the revised manuscript and further improve the presentation of our motivation.
>
>
> - **Discussion of limitations**
>
>     Thank you for raising this point. We acknowledge two primary limitations of SpecExit. First, as an early-exit method, SpecExit is effective mainly when the base model produces redundant reasoning for a given problem. For tasks at the boundary of the model’s capability, the opportunity for early termination is limited. Second, integrating SpecExit into production serving frameworks such as vLLM may require additional engineering effort, due to the need to interact with the speculative decoding process. We will include a dedicated discussion of these limitations in the revised paper.

---

> > ### Author Rebuttal · Reviewer_zyU4 · 2026-04-02
> >
> > Thanks to the authors for their response. However, most of my concerns have not been resolved, and I will maintain my score.

---

> > > ### Author Response · Authors · 2026-04-02
> > >
> > > Thank you for your feedback and for continuing the discussion.
> > >
> > > We noticed that you mentioned having follow-up questions. Could you kindly specify them? We would like to better understand your remaining concerns and address them thoroughly in our final response to ensure the key aspects of SpecExit are clearly presented.
> > >
> > > Sincerely,
> > > Authors

---

### Official Review · Reviewer_1zjd · 2026-03-24

**Soundness:** 2
**Presentation:** 3
**Significance:** 3
**Originality:** 2
**Overall Recommendation:** 4
**Confidence:** 3

**Summary:**

This paper presents SpecExit, a framework for reducing overthinking and inference latency in large reasoning models. The main idea is to use hidden states from a lightweight draft model in a speculative decoding setting to predict both future tokens and whether reasoning can stop early, avoiding the probing overhead used by prior early-exit approaches. The method does not modify the target model and aims to preserve accuracy while shortening chain-of-thought generation.

The authors evaluate SpecExit on several reasoning benchmarks across math, science, logic, and coding tasks. Results demonstrate reduction in generation length and E2E latency, with little or no accuracy loss compared with baselines.

**Compliance With Llm Reviewing Policy:**

Affirmed.

**Final Justification:**

The authors’ rebuttal didn’t completely resolve my concerns on the generalizability and scenario/task limitation of this approach. I will rise the score a bit due to their efforts and clarification but I am not fully convinced on the practical robustness. It would be better to have more detailed studies on scenarios and failure cases.

**Key Questions For Authors:**

1. why not compare with closed-source reasoning models such as OpenAI’s o-series or Anthropic’s Claude reasoning models?
2. what is the theory behind the generalization of your approach?

**Limitations:**

The author did not discuss the limitation.
It is critical to enhance the experiments by including more comprehensive backbone models and using more recent benchmarks as I commented in the Strengths and Weaknesses section.
Also, a more solid argument of why MTL would improve efficiency (which is counter intuitive) will strengthen the theory part.

**Strengths And Weaknesses:**

Strengths:
The paper addresses an important practical problem: large reasoning models often generate unnecessarily overthought, which increases both latency and cost. The main idea, combining speculative decoding with early exit, is practically motivated, and the paper includes several ablations that help explain the contribution of different signals and design choices. The reported efficiency gains could be useful in deployment settings if the method is robust.
It is also reproducible given the repo link.

Weaknesses:
The experimental evidence is not strong enough to fully support its broader claims. The evaluation is limited to a small set of model families, and does not include frontier closed-source reasoning models such as GPT o series or Claude reasoning models, both of which are described by their developers as strong reasoning capable models on challenging benchmarks and reasoning tasks. it would be helpful to test the method on harder, more recent benchmarks such as FrontierMath for advanced mathematical reasoning, LiveCodeBench for contamination-resistant coding evaluation, and SWE-bench Verified for real-world software engineering tasks.
Also, the method is actually a combination of existing methods with slight modification on MTP and MTL. It is also surprising to me that a heavy MTL framework are not costing more cost when using gradient approach with dynamic weighting (which is one of the traditional task balancing approaches) so as to converge to optimum, which lack theoretical support as well.
So, it is difficult to assess whether the proposed method would outperform the frontier reasoning models in terms of latency and precision. It shows more of a piece of engineering work for a limited models than a broader approach to all LRM models.

---

> ### Author Rebuttal · Authors · 2026-03-30
>
> We thank the reviewer for the insightful and valuable comments. We respond to each point below.
>
> - **Q1: Closed-source models**
>
>     SpecExit is designed for local deployment scenarios where practitioners serve open weight LRMs on their own hardware and aim to reduce inference latency. Our method requires **training** a lightweight draft model to **extract signals** that reflect the sufficiency of reasoning from the hidden states of the target model, which is infeasible for closed-source APIs because their internal representations are inaccessible.
>
>     Moreover, comparing end-to-end latency with API-served models would be confounded by network round-trip time, provider-side batching, and undisclosed serving optimizations, making such a comparison neither fair nor informative. We therefore focus on open-weight models that reflect realistic self-hosted deployment, where SpecExit's latency and GPU-hour savings can be measured precisely.
>
> - **Q2: Theory behind generalization**
>
>     We acknowledge that a complete formal proof remains future work. However, we have conducted substantial empirical analyses to support the reliability of hidden-state signals:
>
>     - **Clustering analysis** (Appendix A.1, Figures 7): PCA and UMAP visualizations reveal that hidden states form distinct clusters corresponding to reasoning stages and paragraph-level confidence. This suggests that intermediate hidden states encode information of reasoning sufficiency.
>
>     - **Distribution analysis** (Appendix A.1, Figure 8): The distributions of predicted signal values at stopping positions differ significantly between correct and incorrect final answers, demonstrating discriminative power.
>
>     - **Case studies** (Appendix A.3, Figures 12–13): Qualitative analyses on problems of varying difficulty show that the three signals capture complementary aspects of reasoning sufficiency.
>
>     These observations provide empirical evidence that hidden states encode reasoning-relevant information, motivating their use as early-exit proxies.
>
> - **Weakness: Training cost and inference efficiency**
>
>     We would like to clarify that the multi-task learning and dynamic weighting are applied **only during training**, while the inference speedup comes from **early-exit of redundant reasoning**.
>
>     **Additional training cost is minimal.** Our early-exit supervision is added on top of the existing EAGLE3 speculative decoding training and can be trained jointly at low marginal cost. The auxiliary exit-signal heads consist of a single linear projection $W \in \mathbb{R}^{d \times 3}$ applied to the hidden state $x \in \mathbb{R}^{s \times d}$. We quantify the additional FLOPs as follows:
>
>     *Forward pass*: the linear projection $y = xW$ costs $6sd$ FLOPs; the loss computation (Sigmoid, MSE, MSLE) costs $\approx 15s$ FLOPs.
>
>     *Backward pass*: weight gradient $dL/dW = x^T (dL/dy)$ costs $6sd$ FLOPs; input gradient $dL/dx = (dL/dy) W^T$ costs $6sd$ FLOPs.
>
>     Thus, the total additional cost is $\approx 18sd + O(s)$ FLOPs per sequence, which constitutes a negligible fraction of the draft model's forward pass. Training a full epoch of the joint SpecExit module (EAGLE3 + exit heads) on 5,000 examples requires approximately 1.5 h on 8xH20 GPUs. The exit-signal heads typically converge within 1-3 epochs. We will incorporate these analyses into the revised manuscript.
>
>     **Inference benefit is substantial.** At inference time, the efficiency gain comes entirely from **terminating redundant reasoning early**. Furthermore, the early-exit prediction module can be fused with the FC layer in speculative decoding, enabling feature reuse and near-zero overhead. SpecExit achieves up to 66% reduction in reasoning length and 2.5$\times$ end-to-end speedup. Therefore, the training overhead is far outweighed by the substantial runtime savings.
>
> - **Weakness: Harder benchmarks**
>
>     We note that SpecExit is an **early-exit method** that accelerates inference when the model *can solve the problem but produces redundant reflection*. For tasks that exceed the base model's capability, early-exit cannot help since the model needs every reasoning step it can produce. Our evaluation deliberately covers a range of difficulty levels (from GSM8K to AIME) to demonstrate that SpecExit adapts its exit behavior to task difficulty as shown in Table 1.
>
> - **Weakness: Novelty**
>
>     While our approach conceptually builds upon standard MTP and MTL techniques, the core contribution is the discovery that hidden states inside speculative decoding encode reasoning sufficiency, enabling near zero-overhead early exit. Prior early-exit work such as DEER reduces token count but increases end-to-end latency (Figure 1a) due to additional probing overhead. SpecExit is the first framework to achieve simultaneous token reduction AND end-to-end latency reduction, yielding a 2.5× speedup over EAGLE3. This establishes a novel architectural operating point rather than merely an engineering combination.

---

> > ### Author Rebuttal · Reviewer_1zjd · 2026-04-06
> >
> > Thank you for the detailed response; most points are clarified. However, the benchmark concern remains: the paper does not clearly justify why these specific benchmarks were selected or what task properties make them appropriate for the method. More importantly, it is still unclear when reasoning is actually redundant, what scenarios the method may hurt, and how these cases are identified, especially given the inherently stochastic nature of LLM outputs. Without clarifying these conditions, it is difficult to assess whether the benchmark comparisons are fair or whether the observed gains are robust beyond a fixed model and evaluation setup.

---

> > > ### Author Response · Authors · 2026-04-07
> > >
> > > Thank you for the constructive follow-up. We appreciate the opportunity to further clarify the benchmark selection, redundancy case, failure mode and generalizability of our paper.
> > >
> > > **1. Justification of Benchmark Selection**
> > >
> > > This traces back to the core motivation of `SpecExit`: while we observe long COT would enhance model performance in reasoning tasks, models often produce unnecessarily verbose outputs, framed as `overthinking` problem in our paper introduction. The six benchmarks across math, coding, science and logic are representative of reasoning tasks. They are precisely the domains where the `overthinking` problem is most severe. We did not choose these benchmarks to favor our method; we chose them because they represent the exact reasoning workloads we aim to accelerate.
> > > Regarding the harder benchmarks you mentioned, the base models in our setup yield poor accuracy on them. Since `SpecExit` fundamentally aims to preserve original generation quality, evaluating latency reduction on tasks the base model struggles to solve becomes uninformative for assessing our impact.
> > >
> > > **2. When is Reasoning Actually "Redundant"?**
> > >
> > > We have provided a case study of redundant reasoning in the links below. For an algebraic math problem, although the baseline output eventually reaches the correct answer, it consists of numerous **"Wait, hold on..." self-reflection and self-doubt steps** that total a **7433-token generation length**. In contrast, our method is able to identify the reasoning progress as sufficient and stop the CoT early, yielding a final output of **770 tokens**—achieving an **89.6% saving** on output length.
> > > - [SpecExit Accelerated Output (770 tokens)](https://anonymous.4open.science/r/SpecExit-B802/case_study/math500-intermediate_algebra-1555/specexit-signal_progress.pdf)
> > > - [Baseline Output (7433 tokens)](https://anonymous.4open.science/r/SpecExit-B802/case_study/math500-intermediate_algebra-623/origin-response.pdf)
> > >
> > > **3. Sensitivity to Task Difficulty and Failure Modes**
> > >
> > > `SpecExit` dynamically adapts the volume of redundancy it reduces. For structurally simpler datasets (e.g., GSM8K), the method exits aggressively to maximize token savings. However, on more complex tasks (such as AIME), the method inherently becomes much more conservative, reducing significantly fewer tokens. We attribute this to our core finding: the signal for answer confidence is inherently encoded within the model's hidden states, allowing our exit mechanism to organically adjust to the problem's underlying complexity.
> > >
> > > Regarding the cases where our method might negatively impact performance, we have provided a detailed categorization of failure modes. **Please kindly refer to our response to Reviewer oHYv (Weakness 2)** for a case study of when and why `SpecExit` may fail.
> > >
> > > **4. Robustness Beyond a Fixed Evaluation Setup**
> > >
> > > Our observed gains are not artifacts of a fixed model setup. We proved this by evaluating across 3 different model architectures and 6 diverse datasets, achieving universal e2e speedups using the same thresholds without dataset-specific tuning.
> > >
> > > We thank you again for pushing for these clarifications. We commit to integrating this detailed discussion on benchmark selection, reasoning redundancy, and method robustness into our revised version. We hope this reply resolves your remaining concerns, and we kindly ask you to consider updating your evaluation score if you find our clarifications satisfactory.

---

### Decision · Program_Chairs · 2026-04-30

**Decision:**

Accept (regular)

**Comment:**

This paper received generally positive but somewhat mixed feedback, with reviewers recognizing several notable strengths:

- It addresses an important and practical problem—reducing overthinking and inference latency in large reasoning models—through a clean and well-motivated design that integrates early-exit signals into speculative decoding without modifying the target model.
- The approach is elegant and efficient, leveraging draft-model hidden states with minimal overhead, and supported by thorough ablations that clarify the contribution of different components.
- Empirical results are strong, showing consistent reductions in token usage and end-to-end latency across multiple benchmarks while largely preserving accuracy.

At the same time, reviewers noted some limitations and some reviewers expressed that their concerns were not fully addressed, albeit with weak accept ratings. The evaluation could be broadened to include stronger baselines, more diverse tasks, and larger or frontier models to better establish generality. In addition, aspects such as failure mode analysis, signal reliability, and training or deployment costs would benefit from deeper investigation to strengthen confidence in robustness and applicability.

Overall, the paper presents a useful and practically relevant contribution with solid empirical gains, though some questions remain about generalization and completeness of analysis. Therefore, I recommend weak acceptance.